# Hippocampal-hypothalamic circuit controls context-dependent innate defensive responses

Jee Yoon Bang[1], Julia Kathryn Sunstrum[2], Danielle Garand[1], Gustavo Morrone Parfitt[3,4], Melanie Woodin[1], Wataru Inoue[2], Junchul Kim[1,4]*

[1]Cell and Systems Biology, University of Toronto, Ontario, Canada; [2]Robarts Research Institute, Western University, Ontario, Canada; [3]Nash Family Department of Neuroscience, Icahn School of Medicine at Mount Sinai, New York, United States; [4]Psychology, University of Toronto, Ontario, Canada

**Abstract** Preys use their memory – where they sensed a predatory threat and whether a safe shelter is nearby – to dynamically control their survival instinct to avoid harm and reach safety. However, it remains unknown which brain regions are involved, and how such top-down control of innate behavior is implemented at the circuit level. Here, using adult male mice, we show that the anterior hypothalamic nucleus (AHN) is best positioned to control this task as an exclusive target of the hippocampus (HPC) within the medial hypothalamic defense system. Selective optogenetic stimulation and inhibition of hippocampal inputs to the AHN revealed that the HPC→AHN pathway not only mediates the contextual memory of predator threats but also controls the goal-directed escape by transmitting information about the surrounding environment. These results reveal a new mechanism for experience-dependent, top-down control of innate defensive behaviors.

## Editor's evaluation

In this study, the authors provide novel insights into the mechanisms by which the brain computes contextual information associated with innate threats in mice. Specifically, it provides the first causal evidence of a hippocampus-anterior hypothalamic pathway mediating spatial fear memory of ethological threats. Overall, these findings should interest a broad scientific audience.

*For correspondence: junchul.kim@utoronto.ca

## Introduction

Manoeuvring through a rapidly changing environment while avoiding the threat of predation is essential for the survival and reproduction of all species (*Anderson and Perona, 2014*). This requires abilities to perceive the magnitude of predator threats (i.e. stimulus detection and integration), initiate defensive responses such as escape flight or freezing (i.e. defensive motor actions), and in parallel, remember the area where the predator appeared (i.e. memorization) so that the possibility of re-encountering the same threat can be avoided (*Gross and Canteras, 2012*; *Silva et al., 2016a*). Upon detecting predatory threats, prey animals also select the most successful defense strategy based on their knowledge of the surrounding environment such as the presence of nearby food and the availability of a safe shelter (*Evans et al., 2019*; *Cooper, 2019*). For example, when there is no safe shelter, rodents select freezing over escape flight to avoid being detected by predators. Once they learn about the existence of a safe shelter, however, defense strategy quickly switches to escape-running toward the shelter (*Vale et al., 2017*; *Evans et al., 2018*). Thus, defensive response to predatory

threats is not simple stimulus-response, but a flexible, cognitive process that utilizes the knowledge of prior experiences and environments (*Evans et al., 2019*; *Vale et al., 2017*; *Blanchard, 2018*).

Innate defensive behaviors are generated by the medial hypothalamic defensive system (*Canteras, 2002*), consisting of the anterior hypothalamic nucleus (AHN) (*Fuchs et al., 1985*; *Lamontagne et al., 2016*), the dorsomedial and central region of the ventromedial hypothalamus (VMHdm/c) *Fuchs et al., 1985*; *Wang et al., 2015*; *Silva et al., 2016b*; *Pérez-Gómez et al., 2015* and the dorsal premammillary nucleus (PMD) (*Canteras, 2002*; *Canteras, 2008*; *Wang et al., 2021a*; *Wang et al., 2021b*). These three distinct nuclei are densely interconnected and become highly active upon predator exposure (*Dielenberg et al., 2001*; *Blanchard et al., 2005*; *Martinez et al., 2008*; *Mendes-Gomes et al., 2020*) to control motor outputs at the level of periaqueductal gray (PAG) (*Evans et al., 2018*; *Wang et al., 2015*; *Canteras and Goto, 1999*; *Tovote et al., 2016*). In both rodents and non-human primates, direct stimulation of the medial hypothalamic defensive system evokes strong defensive responses, such as escape flight, freezing, sympathetic activation, and panic, while its inhibition reduces defensive responses to predator threats (*Silva et al., 2016b*; *Lammers et al., 1988*; *Lipp and Hunsperger, 1978*; *Siegel and Pott, 1988*).

How are then the hard-wired defensive responses flexibly controlled by animals' memory and knowledge of the environment? While the medial hypothalamus defense system has been extensively studied, it remains unknown how information about threat-associated context and spatial environment is implemented at the circuit level during the innate defensive response to predator threats. It is well-established that the environmental context of a salient event is first encoded within the hippocampus (HPC) as the collective activity of place cells and time cells (*Gross and Canteras, 2012*; *Silva et al., 2016a*; *Maren et al., 2013*; *Pentkowski et al., 2006*; *Kjelstrup et al., 2002*; *Wang et al., 2013*; *Lisman et al., 2017*). Later during memory recall, the contextual information serves as a potent retrieval cue by reinstating patterns of brain activity observed during the original experience. (*Aqrabawi and Kim, 2018*; *Mandairon et al., 2014*; *Gottfried et al., 2004*).

Given the critical role of the hippocampus in encoding contextual memory, we hypothesized that hippocampal inputs to the medial hypothalamic defensive system may control innate defensive responses based on the animals' knowledge of the surrounding environment. Using a combination of anterograde tracing and electrophysiological recording, we first found that the hippocampus innervates almost exclusively the AHN within the medial hypothalamic defensive system (i.e., HPC→AHN pathway), but not the PMD or VMHdm/c. Subsequent optogenetic activation and inhibition experiments showed that the HPC→AHN pathway not only mediates the contextual memory of predator threats but also controls the goal-directed escape by transmitting information about the surrounding environment.

## Results

### AHN stimulation evokes escape responses

To examine the behavioral consequences of anterior hypothalamic nucleus (AHN) activation, we transduced neurons in the AHN by bilateral injection of adeno-associated viral vector (AAV) with human synapsin promoter (hSyn) carrying channelrhodopsin-2 (AAV-hSyn-ChR2-eYFP) or AAV-CB7-CI-eGFP for GFP controls (*Figure 1a and b*). The location of viral transduction and optic fiber placement were confirmed to be in the central and caudal regions of AHN with minimal spread to neighbouring hypothalamic areas (*Figure 1—figure supplement 1*). We first examined the effects of low- and high-frequency (6 Hz and 20 Hz) stimulation and found that the high-frequency stimulation generated robust behavioral responses in the absence of any overt predator threat, including jumping, freezing, and running, whereas the low-frequency stimulation increased only freezing (*Figure 1—figure supplement 2*). To systematically investigate the behavioral effects of AHN stimulation, we optogenetically stimulated the AHN in three different escape conditions with varying degrees of difficulty (*Figure 1c and d*): (1) an open field arena with short transparent walls (condition 1, easy), tall opaque walls (condition 2, hard), and physical restraint tube (condition 3, impossible). In condition 1, AHN stimulation induced bursts of running ( > 0.3 m/s) with a short latency (5 ± 1.29 s) (*Video 1*). After bouts of running, AHN-ChR2 mice, but not GFP-control mice, initiated multiple escape jumps which resulted in five of six AHN-ChR2 mice escaping the test arena. We quantified the light-induced behavioral effect as a normalized difference between baseline epoch (OFF, 2 min) and stimulation epoch (ON, 2 min)

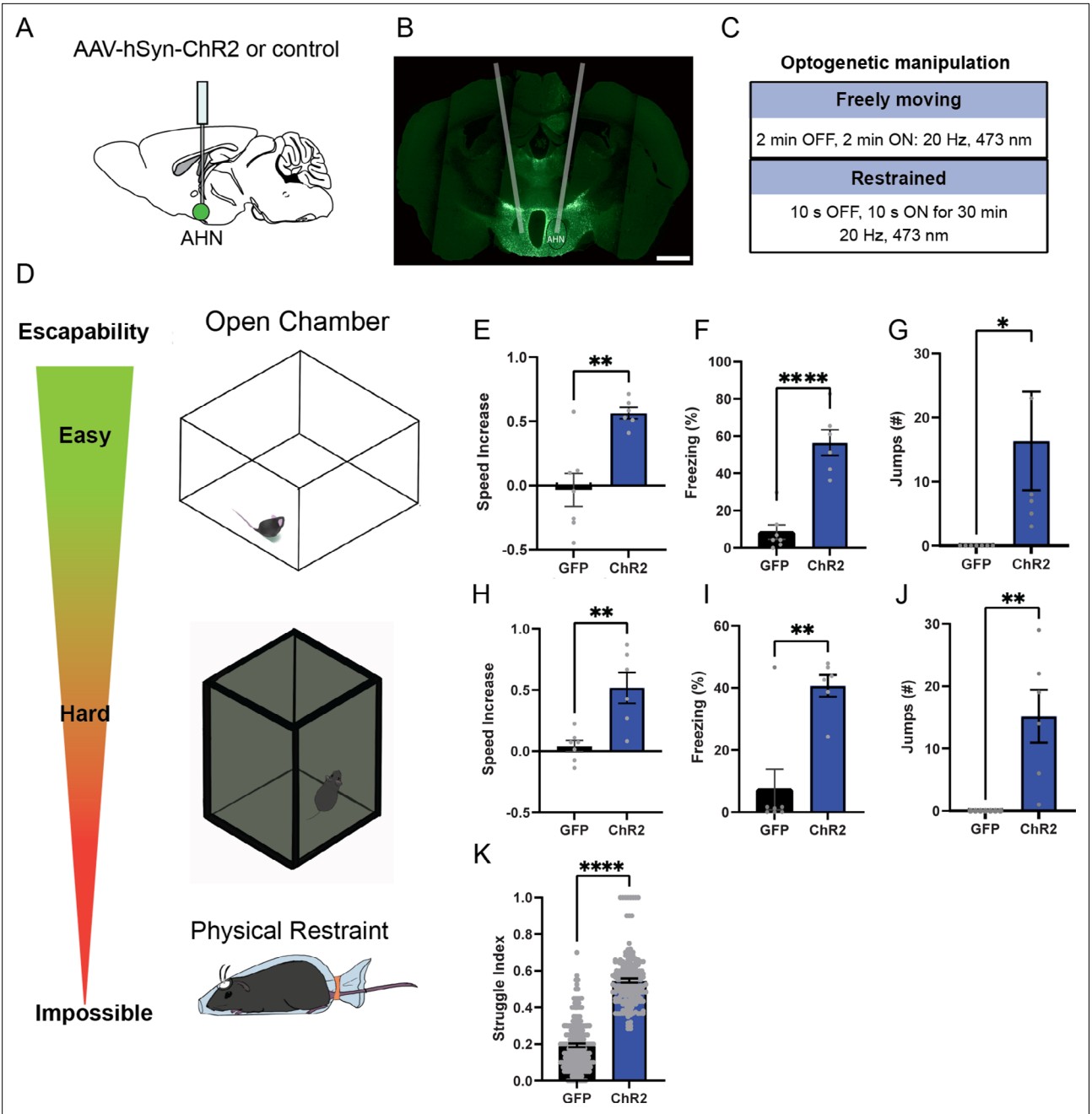

**Figure 1.** AHN stimulation induces escape-associated behaviors. (**a**) Schematic illustration of optogenetic activation in the AHN (green circle depicts the AAV infusion). (**b**) An example of histological confirmation showing the expression of ChR2 and placement of optic fiber in the AHN. (**c**) Schematic describing optogenetic stimulation paradigm. (**d**) Three different escape conditions where the effects of AHN stimulation was examined. Top: open field arena with short transparent walls (condition 1, easy). Middle: tall opaque walls (condition 2, hard). Bottom: physical restraint tube (condition 3, impossible). (**e**) Condition 1: speed increase from the light OFF epoch to ON epoch (GFP N=7, ChR2 N=6 unpaired t-test, two-tailed, t=4.119, df=11, **p=0.0017). (**f**) Condition 1: freezing time during the light ON epoch (GFP N=7, ChR2 N=6, unpaired t-test, two-tailed, t=6.695, df=11, ****p<0.0001). (**g**) Condition 1: number of jumps during the light ON epoch (unpaired t-test, two-tailed, t=2.308, df=11, *p=0.0414). (**h**) Condition 2: speed increase from the light OFF epoch to ON epoch (GFP N=7, ChR2 N=6, unpaired t-test, two-tailed, t=3.778, df=11, **p=0.0031). (**i**) Condition 2: freezing time during the light ON epoch (GFP N=7, ChR2 N=6, unpaired t-test, two-tailed, t=4.259, df=11, **p=0.0013). (**j**) Condition 2: number of jumps during the light ON epoch (GFP N=7, ChR2 N=6, unpaired t-test, two-tailed, t=3.796, df=11, **p=0.003). (**k**) Condition 3: struggle movement during the 30 min of physical restraint (GFP N=4, ChR2 N=6, unpaired t-test, two-tailed, t=26.05, df=366, ****p<0.0001). All results reported are mean ± s.e.m. *p < 0.05, **p < 0.01, ***p < 0.001, ****p<0.0001. Scale bar=1 mm.

The online version of this article includes the following source data and figure supplement(s) for figure 1:

*Figure 1 continued on next page*

*Figure 1 continued*

**Source data 1.** Numerical data shown in *Figure 1*.

**Figure supplement 1.** Compilation of viral expression and optic fiber implantation sites for optogenetic manipulation experiments.

**Figure supplement 2.** The effects of low vs high-frequency AHN stimulation.

**Figure supplement 2—source data 1.** Numerical data shown in *Figure 1—figure supplement 2*.

and found that AHN-ChR2 mice had significant increases in the speed of locomotion, freezing, and jumping (*Figure 1e–g*) compared to GFP controls. In condition 2 (hard), no AHN-ChR2 mice escaped the test arena, but escape attempts were maintained with increased running, freezing, and jumping compared to GFP controls (*Figure 1h–j*, *Video 2*). In condition 3, animals were physically restrained and received AHN stimulation (10 s ON, 10 s OFF) for 30 min, during which escape struggle movements were visually inspected and monitored using a collar sensor with a pulse oximeter. Despite limited mobility and the long duration of physical restraint, AHN-ChR2 mice, but not GFP controls, displayed persistent escape-struggle movements throughout the test. Thus, our data demonstrate that AHN activity is sufficient to evoke escape-associated behavioral responses in the absence of overt predator cues.

## AHN activation carries negative valence and induces conditioned avoidance

Fear of predators is an aversive emotional state that elicits defensive behaviors such as freezing and escape flight (*Wang et al., 2021a*; *Gottfried et al., 2004*; *Clinchy et al., 2013*). Therefore, we probed the emotional valence of AHN activation in a close loop real-time place avoidance assay (RTPA) (*Figure 2a and b*). During a 5-min habituation, mice were allowed to explore two distinct chambers, and a preferred chamber was selected as the photostimulation chamber (*Figure 2c and d*). During a subsequent 20-min RTPA test, mice explored two chambers and received AHN stimulation at either low or high frequency (6 or 20 Hz) only in the photostimulation chamber (*Figure 2b*). All AHN-ChR2 mice exhibited dramatic flight responses upon AHN activation, immediately leaving and avoiding the photostimulation chamber (*Figure 2e*, *Video 3*). While both 6 and 20 Hz stimulation induced significant avoidance of the photostimulation chamber in ChR2 animals, there was a frequency-dependent magnitude of aversion. The 20 Hz stimulation induced a greater mean aversion index (- 0.89) than the 6 Hz stimulation (–0.63)

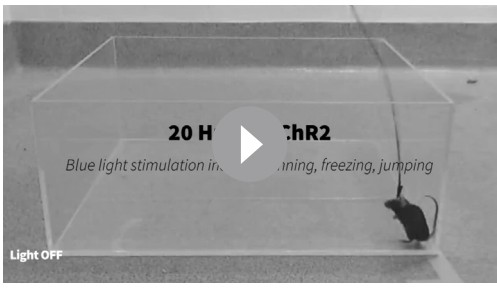

**Video 1.** High-frequency (20 Hz) stimulation of AHN induced running, freezing and jumping in easy escape conditions. During Light OFF, AHN-ChR2 animals are walking, grooming and rearing. Upon 20 Hz light photostimulation, AHN-ChR2 animals display running, freezing and jumping responses in the escapable chamber. In contrast, AHN-GFP animals display no change in behaviors between light OFF and light ON epoch.

https://elifesciences.org/articles/74736/figures#video1

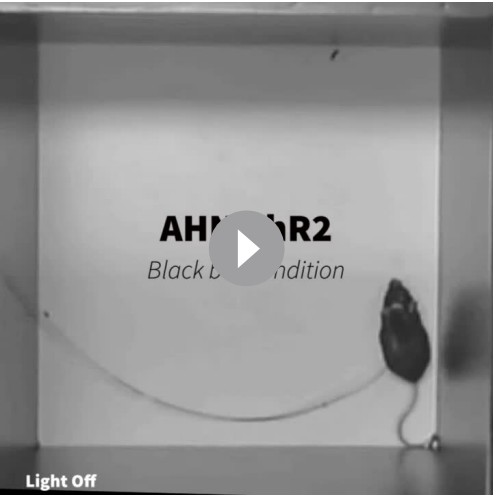

**Video 2.** High-frequency (20 Hz) stimulation of AHN induced running, freezing and jumping in difficult escape conditions. During Light OFF, AHN-ChR2 animals are walking, grooming and rearing. Upon 20 Hz light photostimulation, AHN-ChR2 animals display running, freezing and jumping responses in the inescapable chamber. In contrast, AHN-GFP animals display no change in behaviors between light OFF and light ON epoch.

https://elifesciences.org/articles/74736/figures#video2

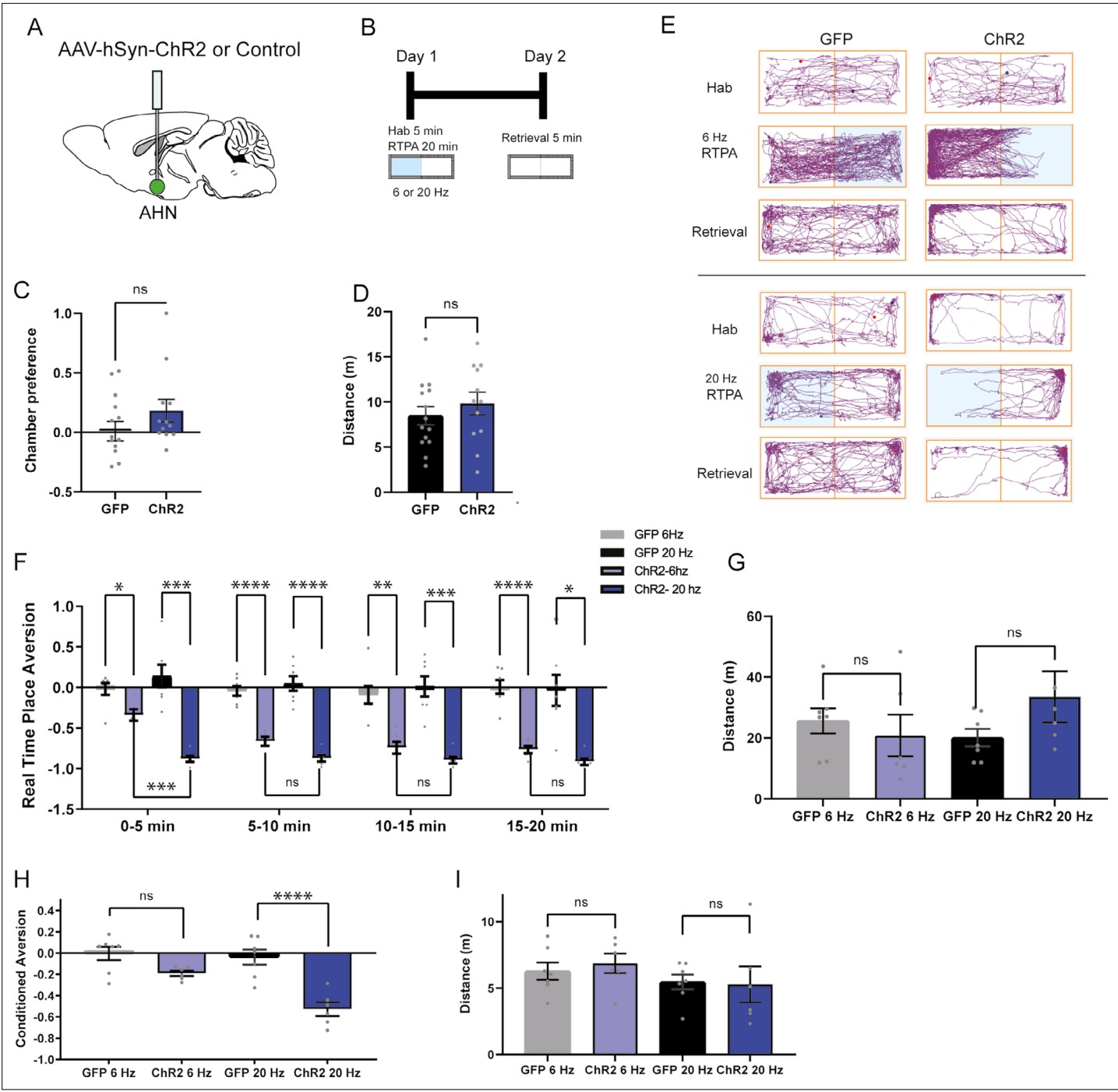

**Figure 2.** AHN stimulation is aversive and induces conditioned place aversion. (**a**) Schematic illustration of optogenetic activation in the AHN (green circle depicts the AAV infusion). (**b**) Schematic describing the RTPA and CPA test paradigm: day 1 consisting of habituation and real-time place preference (20 min) and day 2 for testing conditioned place preference (5 min). (**c**) Chamber preference during habituation (GFP N=14, ChR2 N=12, unpaired t-test, t=1.390, df=24, p=*0.1772,* NS). (**d**) Distance travelled during habituation (GFP N=14, ChR2 N=12, unpaired t-test, t=0.8396, df=24, p=*0.41*, NS). (**e**) Representative locomotion trajectory for a GFP control animal (left column) and a ChR2-expressing animal (right column) during habituation (hab), 6 Hz or 20 Hz real-time stimulation (6 Hz RTPA, 20 Hz RTPA), and conditioned place aversion test (Retrieval). Light-coupled chambers are shown in blue. (**f**) Realtime place aversion monitored across 20-min test (GFP N=7, ChR2 N=6). GFP 6 Hz vs. ChR2 6 Hz (two-way RM ANOVA, time x treatment, F(3,33)=3.965, *\*p=0.016,* time effect, F(2.252, 24.77)=4.739, p=0.152, NS, treatment effect, F(1, 11)=77.41, *\*\*\*\*p<0.0001,* Sidak's multiple comparisons test, 0-5 min, *\*p=0.0359,* 5-10 min, *\*\*\*\*p<0.0001,* 10-15 min, *\*\*p=0.0022,* 15-20 min, *\*\*\*\*p<0.0001*). GFP 20 Hz vs. ChR2 20 Hz (two-way RM ANOVA, time x treatment, F(3,33)=0.6059, p=0.6158, NS, time effect, F(1.938, 21.32)=1.305, p=0.2911, NS, treatment effect, F(1,11)=43.38, *\*\*\*\*p<0.0001,* 24 multiple comparisons test, 0-5 min, *\*\*\*p=0.0008,* 5-10 min, *\*\*\*\*p<0.0001,* 10-15 min, *\*\*\*p=0.0007,* 15-20 min, *\*p=0.0127*). GFP 6

*Figure 2 continued on next page*

Figure 2 continued

Hz vs. GFP 20 Hz (two-way RM ANOVA, time x frequency, F(2.071, 12.42)=1.076, p=0.3730, NS, time effect, F(1.964, 11.78)=0.5391, p=0.5939, NS, frequency effect, F(1, 6)=0.2474, p=0.6366, NS, Sidak's multiple comparisons test, 0-5 min, p=0.8256, NS, 5-10 min, p=0.8824, NS, 10-15 min, p=0.9794, NS, 15-20 min, p=0.9995, NS). ChR2 6 Hz vs. ChR2 20 Hz (2-WAY RM ANOVA, time x frequency, F(1.455, 7.274)=7.391, *p=0.0223, time effect, F(1.514, 7.571)=11.05, **p=0.0075, frequency effect, F(1, 5)=20.99, **p=0.0059, Sidak's multiple comparisons test, 0-5 min, ***p=0.0008, 5-10 min, p=0.2586, NS, 10–15 min, p=0.5763, NS, 15–20 min, p=0.3504, NS). (**g**) Distance travelled during 6 Hz and 20 Hz real-time stimulation (2-WAY ANOVA, frequency x genotype, F(1,22)=2.581, p=0.1224, NS, frequency effect, F(1, 22) = 0.3967, p=0.5353, NS, genotype effect, F(1, 22)=0.5732, p=0.457, NS, Sidak's multiple comparisons test, 6 Hz GFP vs. ChR2, p=0.8013, NS, 20 Hz GFP vs. ChR2, p=0.2058, NS). (**h**) Conditioned aversion memory tested 24-hr after real time place aversion tests (two-way ANOVA, frequency x genotype, F(1,22)=6.208, *p=0.0207, frequency effect, F(1, 22) = 9.411, **p=0.0056, genotype effect, F(1, 22)=31.19, ****p<0.0001, Sidak's multiple comparisons test, 6 Hz GFP vs. ChR2, p=0.0778, NS, 20 Hz GFP vs. ChR2, ****p<0.0001). (**i**) Distance travelled during the conditioned place aversion test (two-way ANOVA, frequency x genotype, F(1,22)=0.2058, p=0.6545, NS, frequency effect, F(1, 22) = 1.998, p=0.1715, NS, genotype effect, F(1, 22)=0.06095, p=0.8073, NS, Sidak's multiple comparisons test, 6 Hz GFP vs. ChR2, p=0.8596, NS, 20 Hz GFP vs. ChR2, p=0.9868, NS). All results reported are mean ± s.e.m. *p < 0.05, **p < 0.01, ***p < 0.001, ****p < 0.0001.

The online version of this article includes the following source data for figure 2:

**Source data 1.** Numerical data shown in *Figure 2*.

(*Figure 2f*) with no difference in total distance travelled during the test (*Figure 2g*). Since the AHN photostimulation was paired with a distinct context, we next asked whether the aversion evoked by AHN activity is sufficient to induce conditioned place avoidance (CPA). A day after the real-time place aversion test, mice were placed back in the middle of the two chambers, but without photostimulation, to determine their conditioned place aversion. Most AHN-ChR2 mice immediately turned away from the photostimulation chamber and exhibited investigatory behaviors towards the entrance of photostimulation chamber (*Video 3*). Both 6 and 20 Hz stimulation produced significant conditioned place aversions with the 20 Hz stimulation inducing a greater mean aversion index (- 0.53) than the 6 Hz stimulation (- 0.20). Together, our data demonstrate that AHN activity carries negative emotional valence and can serve as a stimulus for the formation of a conditioned place aversion memory.

## AHN receives direct glutamatergic inputs from the hippocampus

Next, we investigated the distribution of the hippocampal fiber afferents to the hypothalamus. To this end, we performed anterograde tracing from the hippocampus using virally delivered-ChR2 (AAV-hSyn-ChR2-eYFP) as an anterograde tracer (*Figure 3a*, *Figure 3—figure supplement 1a*). HPC infusions led to the expression of ChR2-eYFP in the ventral two-thirds of the HPC, with minimal spread into adjacent cortical structures such as the entorhinal cortex which does not project to the AHN (*Figure 3—figure supplement 1b*). Consistent with previous reports, GFP-positive axon terminals were detected in the known targets of HPC, including the amygdala (*Takahashi et al., 2005*; *Parfitt et al., 2017*), lateral septum (*Takahashi et al., 2005*; *Canteras and Swanson, 1992*; *Risold and Swanson, 1997*; *Parfitt et al., 2017*, nucleus accumbens *Amaral and Witter, 1989*; *Friedman et al., 2002*) and prefrontal cortex (*Parfitt et al., 2017*; *Amaral and Witter, 1989*; *Friedman et al., 2002*) (data not shown). Within the hypothalamus, HPC axon terminals were found most abundantly in the AHN based on a normalized measure of GFP fluorescence intensity (*Figure 3b and c*). In stark contrast, the VMHdm/c and PMD, the other two main components of the medial hypothalamic defense system were almost excluded from the HPC innervation. Furthermore, the hippocampal innervation of the AHN showed an overall bias against other medial hypothalamic nuclei implicated in stress-induced corticosterone release (PVN) and social aggression (VMHvl) (*Figure 3b*

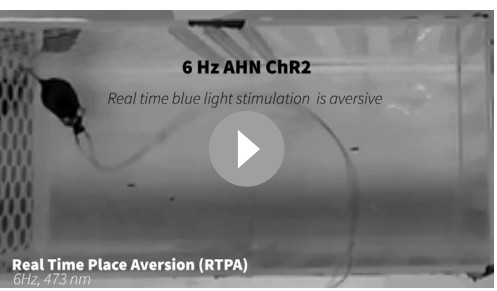

**Video 3.** Low-frequency (6 Hz) stimulation of AHN carries negative valence and induces conditioned avoidance. AHN-ChR2 animals run away from the light-paired chamber when photostimulation is delivered real time. Twenty-four hr later, the same animals remember the negative valence of the light-paired chamber and avoid and escape from the same chamber and remain in the light-off chamber. AHN-GFP animals display no aversion to light-paired chamber real time and 24 hr later.

https://elifesciences.org/articles/74736/figures#video3

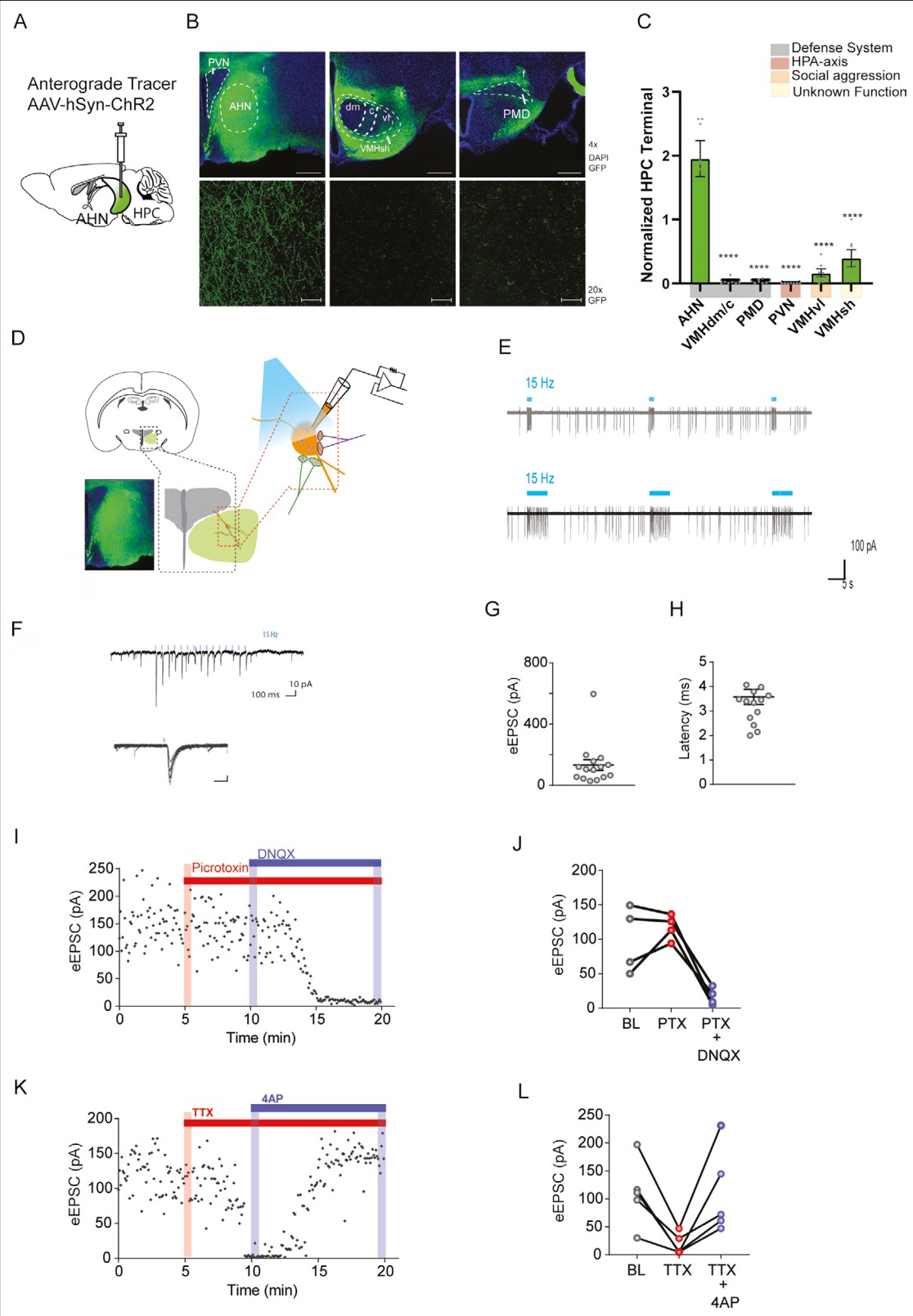

**Figure 3.** Hippocampus sends monosynaptic excitatory inputs to the anterior hypothalamic nucleus. (**a**) Schematic illustration of anterograde tracing experiment. (**b**) HPC terminals (green) in the hypothalamus, including anterior hypothalamic nucleus (AHN), dorsomedial and central regions of ventromedial hypothalamus (VMHdm/c), premammillary dorsal nucleus (PMD), paraventricular nucleus (PVN), ventrolateral region of ventromedial hypothalamus (VMHvl), shell of ventromedial hypothalamus (VMHsh). DAPI staining (blue). (**c**) Quantification of HPC terminal intensity (N=2 animals,

*Figure 3 continued on next page*

*Figure 3 continued*

~7 sections per ROI, One-Way ANOVA, F(5,35)=33.24, ****p<0.0001, Dunnett's multiple comparisons test, AHN vs. VMHdm/c, ****p<0.0001, AHN vs. PMD, ****p<0.0001, AHN vs. PVN, ****p<0.0001, AHN vs. VMHvl, ****p<0.0001, AHN vs. VMHsh, ****p<0.0001). (**d**) Schematic illustration for patch clamp recordings of AHN neurons in coronal brain slices that express ChR2 in HPC terminals. (**e**) Examples of cell attach recordings. Illumination of blue light (480 nm, 5ms pulse at 15 Hz) triggered firing of AHN neurons. (**f**) Examples of whole-cell voltage-clamp recordings of AHN neurons. Blue light illumination (5 ms) evoked inward current. (**g-h**) Summary of light-evoked EPSCs (**g**) amplitude and latency (**h**). (**i**) Light-evoked EPSCs persisted in the presence of GABA A receptor antagonist picrotoxin (PTX, 100 µM) and eliminated by AMPA/kainite receptor antagonist DNQX (20 µM). (**j**) Summary of eEPSC change after PTX and DNQX application. (**k**) Light-evoked EPSCs were eliminated by TTX (0.5 µM) and then recovered by a low dose 4-AP (100 µM). (**l**) Summary of eEPSC changes after TTX and 4-AP application. All results reported are mean ± s.e.m. *p < 0.05, **p < 0.01, ***p < 0.001, and ****p<0.0001. Scale bar=100 µm.

The online version of this article includes the following source data and figure supplement(s) for figure 3:

**Source data 1.** Numerical data shown in *Figure 3*.

**Figure supplement 1.** Viral expression of the anterograde tracer ChR2-eYFP.

**Figure supplement 2.** HPC inputs innervate both GABA and non-GABA cells in the AHN.

**Figure supplement 2—source data 1.** Numerical data shown in *Figure 3—figure supplement 2*.

*and c*). This data indicates that the AHN is the primary entry point for HPC inputs to the medial hypothalamic defense system.

To further validate direct hippocampal inputs arriving at the AHN and determine their electrophysiological properties, we carried out cell-attached and whole cell patch-clamp recordings from AHN cells in acute brain slices containing ChR2-expressing HPC axon terminals (*Figure 3d*). In the cell-attached voltage-clamp mode, photostimulation of HPC terminals (473 nm, 5ms pulses at 15 Hz) triggered robust action potential firings of AHN cells (*Figure 3e*). In the whole-cell voltage clamp mode, photostimulation induced short-latency (average latency 3.6ms) excitatory postsynaptic currents (EPSCs) (average amplitude 132 pA; *Figure 3f–h*). Light-evoked EPSCs in the AHN were not affected by GABAA receptor antagonist picrotoxin (PTX, 100 µM) but eliminated by AMPA/kainite receptor antagonist DNQX (10 µM), indicating that HPC input to the AHN is glutamatergic in nature (*Figure 3i and j*). To isolate monosynaptic inputs from ChR2-expressing HPC axons, we sequentially added tetrodotoxin (TTX, 1 µM) and 4-aminopyridine (4-AP, 100 µM) to the ACSF. The previously observed light-evoked EPSCs were eliminated by TTX but recovered after the application of 4-AP, lending further support that monosynaptic transmission was triggered by direct ChR2-mediated depolarization of HPC terminal boutons (*Hoover and Vertes, 2007*, *Figure 3k and l*).

As AHN is heavily populated by GABAergic cells, we next investigated whether the direct HPC innervation of AHN is biased toward GABA cells. We repeated the current clamp recording experiments with AHN slices from double transgenic reporter mice (RC::Frepe, Dlx5/6-FLPe, *Figure 3—figure supplement 2a*) in which forebrain GABA cells are specifically labeled with red fluorescent protein, mCherry. We observed mCherry labeled GABA cells in the AHN but not in VMHdm/c and PMD (*Figure 3—figure supplement 2b*, bottom row). As expected, photostimulation evoked action potential spikes in mCherry-positive AHN GABA cells. However, we did not find any significant difference in the number of photostimulation-induced spikes between mCherry-positive and mCherry-negative cells, indicating that HPC axon terminals synapse on both GABA and glutamatergic cells in the AHN (*Figure 3—figure supplement 2c-i*).

Together, our anterograde tracing and electrophysiological recording demonstrate that the AHN receives direct monosynaptic excitatory inputs from the HPC. These findings also suggest that the AHN plays a specialized role in the medial hypothalamic defensive system, different from other major components, namely the PMD and VMHdm/c.

## Activation of HPC→AHN pathway induces escape-associated locomotion

The hippocampus sends direct monosynaptic inputs to the AHN, but their behavioral function remains unknown. Thus, we examined if activating HPC→AHN pathway would induce the same behavioral responses seen in the direct AHN soma activation. The HPC was virally transduced with AAV-hSyn-ChR2-eYFP, and optic fibers were bilaterally implanted at the AHN to illuminate HPC axon terminals (*Figure 4a and b*). The viral transduction was confirmed to include all hippocampal presynaptic sources

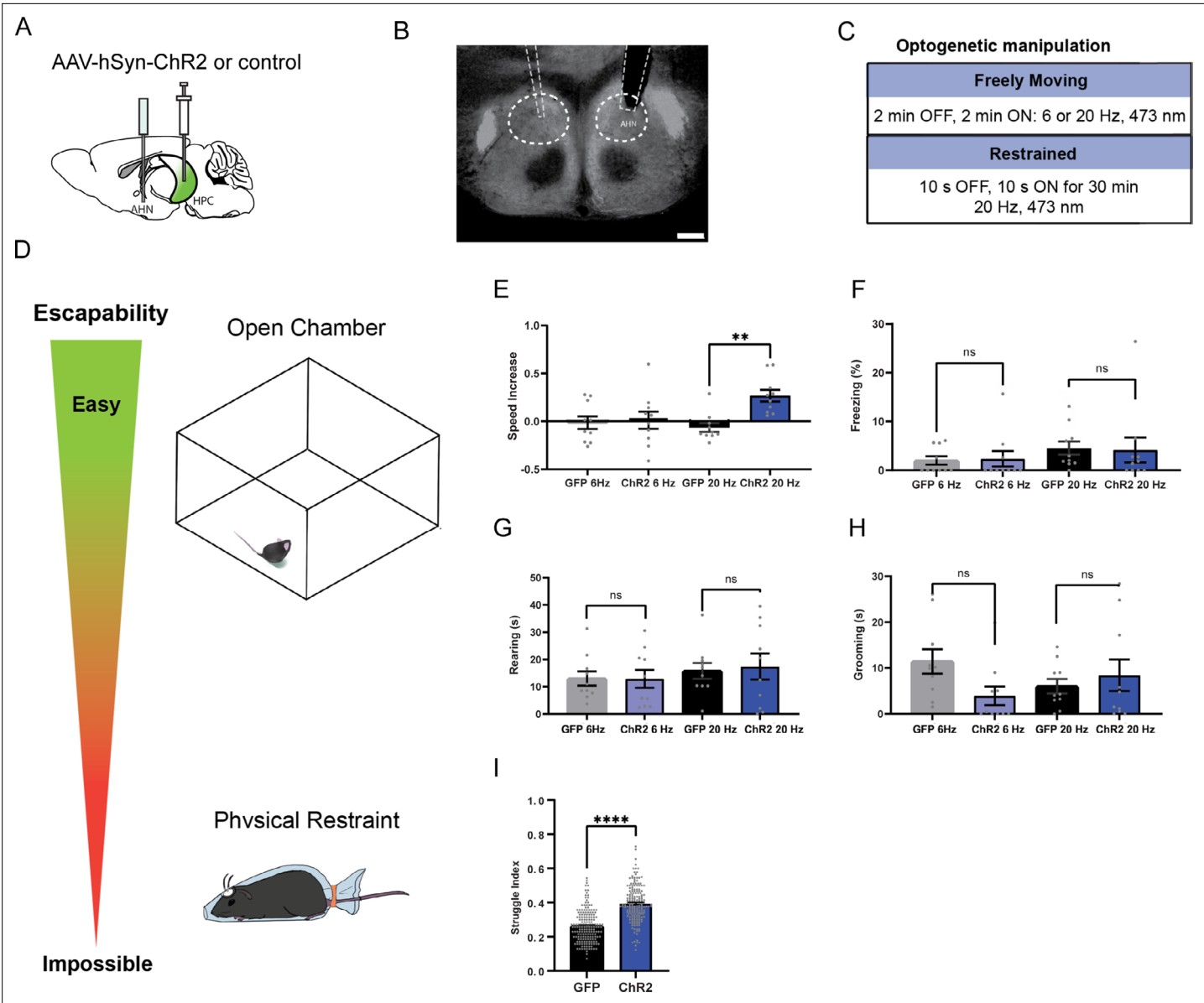

**Figure 4.** HPC→AHN pathway activation induces escape-associated locomotion. (**a**) Schematic illustration of optogenetic activation of hippocampal terminals in the AHN (GFP N=10, ChR2 N=10). (**b**) An example of histological confirmation showing the expression of HPC terminals and placement of optic fibers in the AHN. (**c**) Schematic describing optogenetic stimulation paradigm. (**d**) Two different escape conditions where the effects of HPC terminal stimulation was examined. Top: open field arena with short transparent walls (condition 1, easy). Bottom: physical restraint tube (condition 2, impossible). (**e**) Condition 1: speed increase from the light OFF epoch to ON epoch (two-way ANOVA, frequency x genotype, F(1,36)=5.298,*p=0.0272, frequency effect, F(1, 36)=2.337, p=0.135, NS, genotype, F(1, 36)=7.164, *p=0.0111, Sidak's multiple comparisons test, 6 Hz GFP vs. ChR2, p=0.957, NS, 20 Hz GFP vs. ChR2, **p=0.0024). (**f**) Condition 1: freezing time during the light ON epoch (two-way ANOVA, frequency x genotype, F(1,36)=0.04839, p=0.8273, NS, frequency effect, F(1, 36)=1.637, p=0.2089, NS, genotype effect, F(1, 36) = 2.385e-005, p=0.9961, NS, Sidak's multiple comparisons test, 6 Hz GFP vs. ChR2, p=0.9856, NS, 20 Hz GFP vs. ChR2, p=0.9843, NS). (**g**) Condition 1: rearing time during the light ON epoch (two-way ANOVA, frequency x genotype, F(1,36)=0.06028, p=0.8075, NS, frequency effect, F(1, 36)=1.08, p=0.3057, NS, genotype effect, F(1, 36)=0.04343, p=0.8361, NS, Sidak's multiple comparisons test, 6 Hz GFP vs. ChR2, p=0.9996, NS, 20 Hz GFP vs. ChR2, p=0.9375, NS) (**h**) Condition 1: grooming time during the light ON epoch (two-way ANOVA, frequency x genotype, F(1,36)=3.858, p=0.0573, NS, frequency effect, F(1,36) = 0.03451, p=0.8537, NS, genotype effect, F(1,36)=1.024, p=0.3184, NS, Sidak's multiple comparisons test, 6 Hz GFP vs. ChR2, p=0.083, NS, 20 Hz GFP vs. ChR2, p=0.7549, NS). (**i**) Condition 2: struggle movement during the 30 minutes of physical restraint (GFP N=7, ChR2 N=9, unpaired t-test, two-tailed, t=12.22, df=356 ****p<0.0001). All results reported are mean ± s.e.m. *p < 0.05, **p < 0.01, ***p < 0.001, ****p<0.0001. Scale bar=200μm.

The online version of this article includes the following source data and figure supplement(s) for figure 4:

**Source data 1.** Numerical data shown in *Figure 4*.

*Figure 4 continued on next page*

*Figure 4 continued*

**Figure supplement 1.** viral expression and optic fiber implantation sites for HPC→AHN pathway optogenetic activation a,b, HPC terminal activation in the AHN (N=22).

**Figure supplement 2.** The effects of low- vs high-frequency HPC→AHN pathway activation.

**Figure supplement 2—source data 1.** Numerical data shown in *Figure 4—figure supplement 2*.

of the AHN along the entire dorsoventral axis of the hippocampal formation (dSUB, dCA1, vCA1, vSUB) (*Figure 4—figure supplement 1*). Light induced-behavioral changes were then monitored during low or high frequency (6 or 20 Hz) stimulation of HPC→AHN pathway in an open field arena and compared between ChR2 and GFP control mice (*Figure 4c and d*, *Figure 4—figure supplement 2*). To our surprise, the pathway activation did not elicit robust escape jumping or freezing observed in the direct AHN activation. Instead, it produced light-synched, reversible increases in running bouts and speed (*Figure 4e*, *Figure 4—figure supplement 2c*). We also examined light-induced changes in consummatory (rearing, grooming) behaviors (*Figure 4g and h*, *Figure 4—figure supplement 2*), and found that only grooming was decreased during the 6 Hz light stimulation (*Figure 4f–h*, *Figure 4—figure supplement 2f*). To corroborate the effects of activating HPC→AHN pathway on escape-associated locomotion, we repeated the same pathway activation during physical restraint condition. The delivery of bursts of light pulses (20 Hz, 10 s ON, 10 s OFF) for 30 min significantly increased escape-associated struggle movements in ChR2 mice compared to controls (*Figure 4i*), consistent with the effect of direct AHN soma activation. Thus, our data suggests that HPC→AHN pathway activity promotes escape responses by inducing locomotion.

## HPC→AHN pathway activation is aversive and instructs learning of a conditioned place aversion

To evaluate whether the HPC→AHN pathway activity is intrinsically aversive and sufficient to induce a conditioned place aversion, we used the RTPA and CPA paradigms (*Figure 5a and b*). During the habituation, mice did not show any significant preference to either chamber, and distance travelled did not differ (*Figure 5c and d*). During the subsequent RTPA task, ChR2 mice gradually developed an avoidance to a chamber paired with light stimulation at both 6 Hz and 20 Hz frequency (*Figure 5e and f*, *Videos 4 and 5*). Although there was a trend of increased locomotion with stimulation at 20 Hz frequency, total distance travelled did not differ compared to the controls (*Figure 5g*). One day after the RTPA task, mice were tested for memory retention in the CPA task. ChR2 mice that received HPC→AHN pathway stimulation during RTPA at 6 Hz, but not 20 Hz, displayed a robust conditioned aversion to the stimulation chamber (Figurer 5 h). The distance travelled was not different between controls and ChR2 groups (*Figure 5i*).

## Optogenetic inhibition of HPC→AHN pathway impairs the retrieval of contextual memory of predator cue

The HPC→AHN pathway activity is aversive and can induce a conditioned place aversion. However, the nature of information that the pathway encodes remains unknown. Given the role of the HPC in contextual memory and its direct connection with the AHN, we hypothesized that the HPC→AHN pathway may promote goal-directed escapes by encoding the animal's knowledge or memory of the surrounding environment.

To address this hypothesis, we first investigated the role of HPC→AHN pathway in mediating contextual memory to predatory threats by optogenetically inhibiting the pathway and measuring its effects on conditioned escape responses from an ethologically relevant predator cue. The HPC was virally transduced with AAV-CamKIIa-ArchT-GFP, and optic fibers were bilaterally implanted at the AHN to illuminate HPC axon terminals (*Figure 6a*, *Figure 6—figure supplement 1*). On day 1 (pre-conditioning), mice were habituated to two neutral but visually distinct contexts in a two-chamber apparatus. On days 2 and 3 (conditioning), a predator cue (10% L-Felinine) was paired with one chamber, and water with the other in a counterbalanced manner (*Figure 6b*). L-Felinine is a putative predator kairomone of a Felidae species (*Voznessenskaya et al., 2016*; *Kvasha et al., 2018*) and was chosen as a predator cue because it induces a robust dose-dependent increase in freezing compared to predator urine samples (*Figure 6—figure supplement 2*). During the conditioning, both

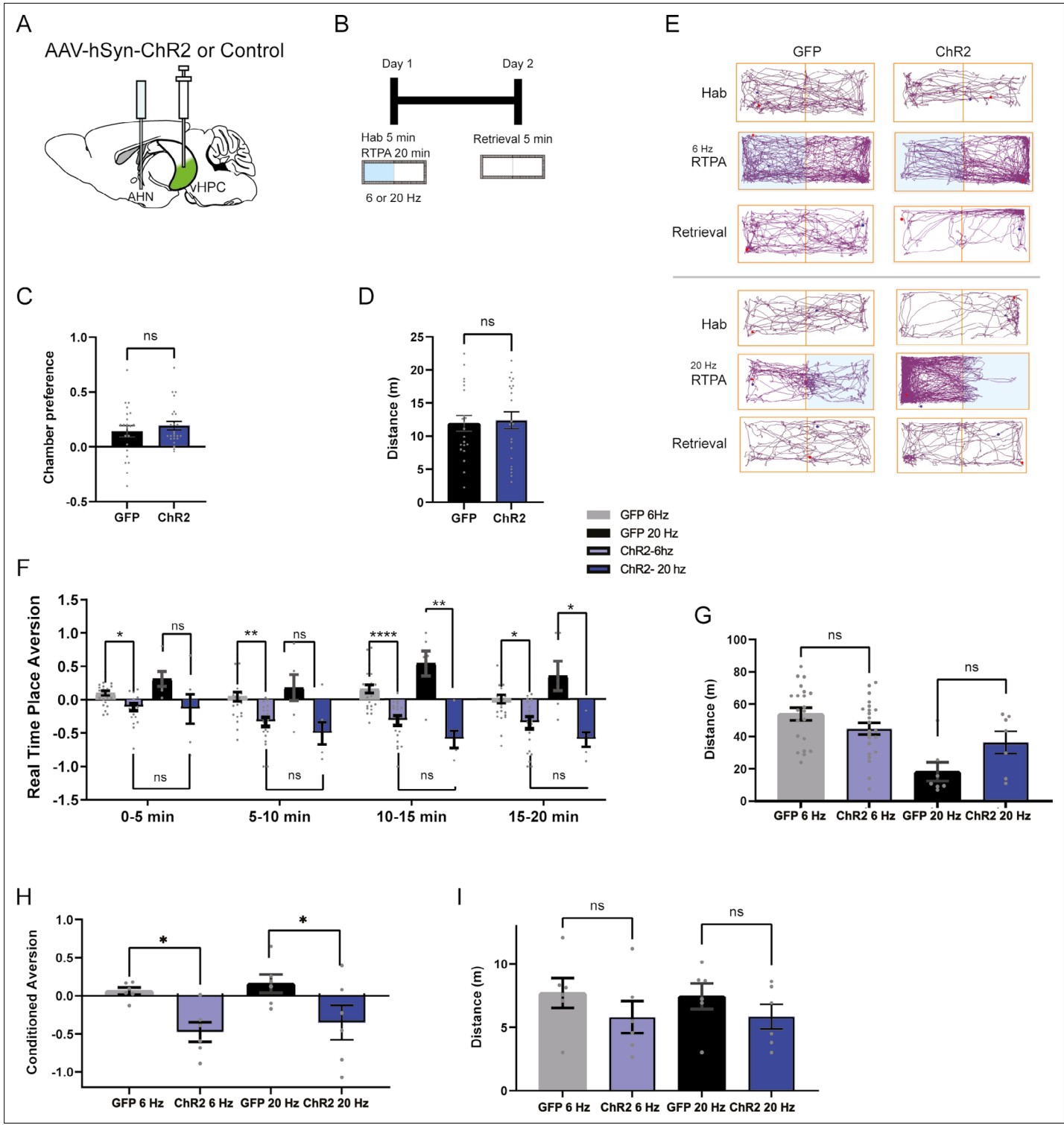

**Figure 5.** HPC→AHN pathway activation is aversive and instructs learning of a conditioned place aversion. (**a**) Schematic illustration of optogenetic activation of hippocampal terminals in the AHN (GFP N=21, ChR2 N=22). (**b**) Schematic describing the RTPA and CPA test paradigm: day 1 consisting of habituation and real-time place preference (20 min) and day 2 for testing conditioned place preference (5 min). (**c**) Chamber preference during habituation (unpaired t-test, two-tailed, t=0.8339, df=41, p=*0.4089*, NS). (**d**) Distance travelled during habituation (unpaired t-test, two-tailed, t=0.2674, df=41, p=*0.7905*, NS). (**e**) Representative locomotion trajectory for a GFP control animal (left column) and a ChR2-expressing animal (right column) during habituation (hab), 6 Hz or 20 Hz real-time stimulation (6 Hz RTPA, 20 Hz RTPA), and conditioned place aversion test (Retrieval). Light-coupled chambers are shown in blue. (**f**) Real time place aversion monitored across 20 minute test. GFP 6 Hz vs. ChR2 6 Hz (two-way RM ANOVA, time x

*Figure 5 continued on next page*

*Figure 5 continued*

treatment F(3,120)=3.539, *p=0.0168, time effect, F(2.633, 105.3)=4.648, **p=0.0062, treatment effect, F(1,40)=21.57, ****p<0.0001, Sidak's multiple comparisons test, 0–5 min, *p=0.0206, 5–10 min, **p=0.0017, 10-15 min, ****p<0.0001, 15-20 min, *p=0.0124), GFP 20 Hz vs. ChR2 20 Hz (two-way RM ANOVA, time x treatment, F(3,30)=4.132, *p=0.0145, time effect, F(2.228, 22.28)=2.056, p=0.1476, NS, treatment effect, F(1,10)=13.59, **p=0.0042, Sidak's multiple comparisons test, 0-5 min, p=0.5916, NS, 5-10 min, p=0.1451, NS, 10-15 min, **p=0.0031,15-20 min, *p=0.0229). GFP 6 Hz vs. GFP 20 Hz (two-way RM ANOVA, time x treatment, F (3, 75) = 1.249, p=0.2980, NS, time effect, F(2.711, 67.77)=3.977, *p=0.0139, treatment effect, F(1,25)=4.669, *p=0.0405, Sidak's multiple comparisons test, 0-5 min, p=0.3668, NS, 5-10 min, p=0.9733, 10-15 min, p=0.396, NS, 15-20 min, p=0.5472, NS) ChR2 6 Hz vs. ChR2 20 Hz (two-way RM ANOVA, time x treatment, F (3, 75) = 1.828, p=0.1492, NS, time effect, F(1.944, 48.61)=10.74, ***p=0.0002, treatment effect, F(1,25) = 1.279, p=0.2687, NS, Sidak's multiple comparisons test, 0-5 min, P=0.9998, NS, 5-10 min, p=0.959, NS, 10-15 min, p=0.3079, NS, 15-20 min, p=0.3884, NS) (**g**) Distance travelled during 6 Hz and 20 Hz real-time stimulation (two-way ANOVA, frequency x treatment, F(1, 51) = 4.679, *p=0.0352, frequency effect, F(1,51)=13.87, ***p=0.0005, treatment effect, F(1,51)=0.2719, p=0.6043, NS, Sidak's multiple comparisons test, 6 Hz GFP vs. ChR2 P=0.1626, NS, 20 Hz GFP vs. ChR2 p=0.252, NS). (**h**) Conditioned aversion memory tested 24-hr after real-time place aversion tests (two-way ANOVA, frequency x treatment, F(1,20)=0.009471, p=0.9234, NS, frequency effect, F(1,20)=0.5755, p=0.4569, NS, treatment effect, F(1,20)=13.05, **p=0.0017, Sidak's multiple comparisons test, 6 Hz GFP vs. ChR2, *p=0.0323, 20 Hz GFP vs. ChR2, *p=0.0433). (**i**) Distance travelled during the conditioned place aversion test (two-way ANOVA, frequency x treatment, F(1, 20)=0.01613, p=0.902, NS, frequency effect, F(1,20)=0.009512, p=0.9233, treatment effect, F(1,20)=2.486, p=0.1305, Sidak's multiple comparisons test, 6 Hz GFP vs. ChR2, p=0.4260, NS, 20 Hz GFP vs. ChR2, p=0.5342, NS). All results reported are mean ± s.e.m. *p < 0.05, **p < 0.01, ***p < 0.001, ****p<0.0001.

The online version of this article includes the following source data for figure 5:

**Source data 1.** Numerical data shown in *Figure 5*.

GFP and ArchT mice displayed increased freezing in the L-Felinine chamber compared to the water chamber (*Figure 6—figure supplement 3*). On Day 4 (post-conditioning), mice were allowed to freely explore the two chambers while the HPC→AHN pathway was optogenetically inhibited (*Figure 6b*). As expected, GFP control mice displayed an avoidance of the L-Felinine context (*Figure 6c and d*, *Figure 6—figure supplement 4a*). ArchT mice, however, failed to remember and avoid L-Felinine context (*Figure 6c and d*, *Figure 6—figure supplement 4b*). Furthermore, the contextual memory impairment was accompanied by significant decreases in defensive behavioral responses such as freezing, escape runs, and grooming compared to GFP control (*Figure 6e–h*, *Video 6*). This finding was replicated in a different CPA paradigm involving 5 days of conditioning, which allowed us to quantify learning of predator context across multiple days (*Figure 6—figure supplement 5a*). Both GFP and ArchT mice developed predator odour context aversion gradually (*Figure 6—figure supplement 5b, c*), and the HPC→AHN pathway inhibition post conditioning resulted in contextual memory impairment (*Figure 6—figure supplement 5d-f*).

Next, we investigated the role of HPC inputs in driving the activities of AHN during the retrieval of contextual memory of predator cues. Nienty min after a post-conditioning test, GFP control and

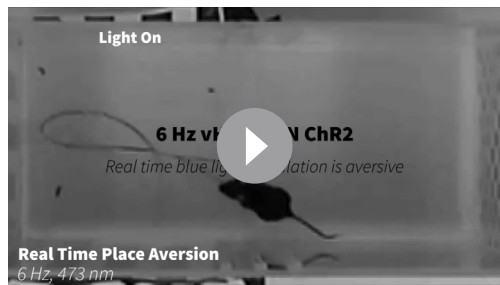

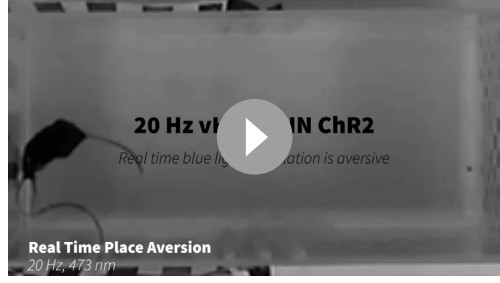

**Video 4.** Low-frequency (6 Hz) stimulation of HPC-AHN carries negative valence and induces conditioned place avoidance. HPC-AHN ChR2 animals run away from the 6 Hz light-paired chamber when photostimulation is delivered real time. Twenty-four hr later, the same animals remember the negative valence of the light-paired chamber and avoid and escape from the same chamber and remain in the light-off chamber. HPC-AHN GFP animals display no aversion to light-paired chamber real time and 24 hr later.

https://elifesciences.org/articles/74736/figures#video4

**Video 5.** High-frequency (20 Hz) stimulation of HPC-AHN carries negative valence and induces conditioned place avoidance. HPC-AHN ChR2 animals run away from the 20 Hz light-paired chamber when photostimulation is delivered real time. Twenty-four hr later, the same animals remember the negative valence of the light-paired chamber and avoid and escape from the same chamber and remain in the light-off chamber. HPC-AHN GFP animals display no aversion to light-paired chamber real time and 24 hr later.

https://elifesciences.org/articles/74736/figures#video5

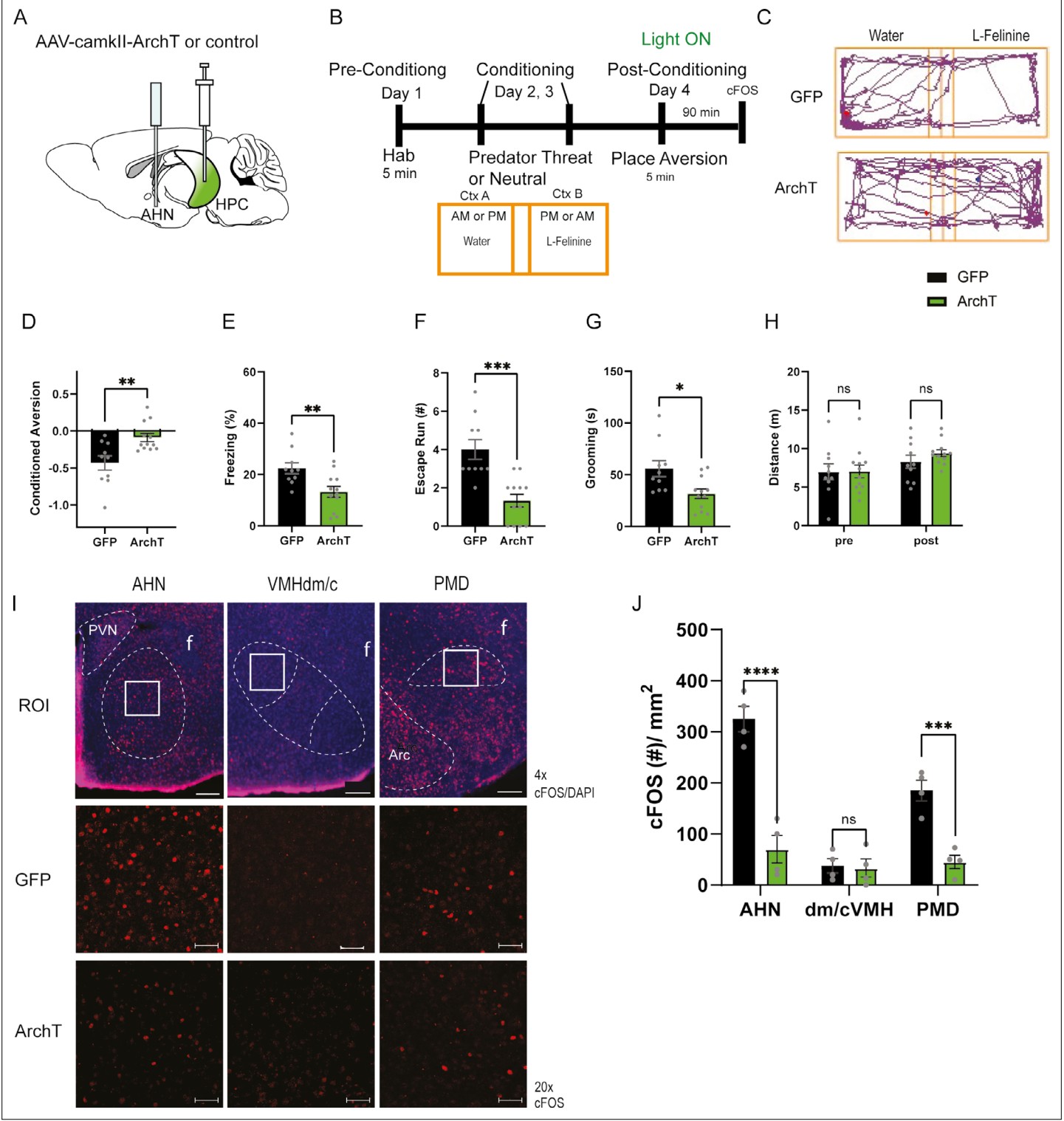

**Figure 6.** HPC input to the AHN is necessary for remembering the context-associated with predatory threat. (**a**) Schematic illustration of optogenetic HPC terminal inhibition in the AHN (GFP N=10, ArchT N=12). (**b**) Schematic describing the behavioral paradigm for the contextual fear conditioning with the predator odor (L-Felinine-); day 1 for habituation (5 min), days 2–3 for two daily conditioning sessions where mice were enclosed either L-Felinine- or water-paired chamber for 20 minutes in AM and PM in a counterbalanced manner, and day 4 for testing conditioned place preference (5 min) and the immunochemical detection of c-Fos. (**c**) Representative locomotion trajectory for a GFP control animal (top) and a ChR2-expressing animal (down) during the conditioned place aversion test with optogenetic HPC terminal inhibition in the AHN (left: water-coupled chamber, right: L-Felinine-coupled chamber). (**d**) Conditioned aversion memory tested 24 hr after conditioning. GFP vs. ArchT (unpaired t-test, t=3.223, df=20.

*Figure 6 continued on next page*

*Figure 6 continued*

**p=*0.0043*). (**e**) Freezing time during the conditioned place aversion test. GFP vs. ArchT (unpaired t-test, t=3.056, df=20, **p=*0.0062*). (**f**) Number of escape runs from the L-Felinine-paired chamber to the water-paired chamber. GFP vs. ArchT (unpaired t-test, t=4.479, df=20, ***p=*0.0002*). (**g**) Time spent grooming during the conditioned place aversion test. GFP vs. ArchT (unpaired t-test, t=2.816, df=20, *p=*0.0107*). (**h**) Distance travelled during habituation (pre) and conditioned place aversion test (post) (two-way ANOVA, training x treatment, F(1, 20)=0.9938, p=*0.3307*, NS, training effect, F(1,20) = 12.29, **p=*0.0022*, treatment effect, F(1,20)=0.3235, p=0.5759, NS). (**i**) c-Fos immunochemical detection across the medial hypothalamic defense system (AHN, VMHdm/c, PMD). First row, representative 4 x epi-fluorescence microscope images of the medial hypothalamic defense system in GFP control mice. The regions of interest (ROI, white squares) within AHN, VMHdm/c, and PMD were imaged by confocal microscopy for cell counting. Second and third row: representative 20 x confocal images of c-Fos signals in AHN, VMHdm/c, and PMD activated by the conditioned place aversion test in GFP and ArchT mice, respectively. (**j**) Density of c-Fos signals in AHN, VMHdm/c, and PMD in GFP control (black) and ArchT mice (green) (N=4 mice for each group; two-way ANOVA, ROI x treatment, F(2, 18)=19.65, ****p<*0.0001*, ROI effect F(2,18)=32.79, ****p<*0.0001*, treatment effect F(1, 18)=66.19, ****p<*0.0001*, Sidak's multiple comparison test, GFP AHN vs ArchT AHN, ****p<*0.0001*, GFP VMHdm/c vs. ArchT dm/cVMH p=0.9984, NS, GFP PMD vs ArchT PMD, ***p=*0.0003*). All results reported are mean ± s.e.m. *p < 0.05, **p < 0.01, ***p < 0.001, ****p<0.0001. Scale bar = 100 μm for 4 x epi-fluorescence microscope images and 10 μm for 20 x confocal images. (PVN, paraventricular nucleus. f, fornix. Arc, arcuate nucleus).

The online version of this article includes the following source data and figure supplement(s) for figure 6:

**Source data 1.** Numerical data shown in *Figure 6*.

**Figure supplement 1.** Viral expression and optic fiber implantation sites for HPC→AHN pathway optogenetic inhibition.

**Figure supplement 2.** L-Felinine increases freezing in a dose-dependent manner.

**Figure supplement 2—source data 1.** Numerical data shown in *Figure 6—figure supplement 2*.

**Figure supplement 3.** Behavioral responses to L-Felinine during conditioning sessions.

**Figure supplement 3—source data 1.** Numerical data shown in *Figure 6—figure supplement 3*.

**Figure supplement 4.** HPC→AHN pathway inhibition impairs the retrieval of place aversion memory.

**Figure supplement 4—source data 1.** Numerical data shown in *Figure 6—figure supplement 4*.

**Figure supplement 5.** Development of predator associated context avoidance and impairment of retrieval of place aversion memory upon HPC-AHN pathway inhibition.

**Figure supplement 5—source data 1.** Numerical data shown in *Figure 6—figure supplement 5*.

ArchT mice were euthanized for immunochemical detection of c-Fos in the medial hypothalamic defense system. We found that c-Fos expression in ArchT mice was decreased in the AHN and PMD, but not in the VMHdm/c, compared to GFP control (*Figure 6i and j*). Together, our data demonstrates that the HPC→AHN pathway enables animals to avoid the environment associated with predators by driving AHN activities during the retrieval of contextual memory of predator cues.

## Optogenetic activation of HPC→AHN pathway evokes goal-directed escapes to shelter

Another important aspect of escape response, other than predatory threats, is the use of shelter as the escape target. Thus, we tested whether the HPC→AHN pathway plays a role in goal-directed escape to a safe shelter. The HPC was virally transduced with AAV-hSyn-ChR2-eYFP, and optic fibers were bilaterally implanted at the AHN (*Figure 7a*). Mice were then placed in the open field arena containing a shelter box to determine whether the optogenetic pathway stimulation leads to an escape flight to the shelter (*Figure 7c*). During the habituation stage, mice were given 5 min to freely explore the arena and exploit the shelter (*Figure 7b*). Both ChR2 and GFP control mice intermittently visited the shelter and spent a comparable amount of time in the shelter (*Figure 7d and e*). During the stimulation stage, the HPC→AHN pathway was stimulated at 6 or 20 Hz frequency when mice were outside the shelter. The pathway stimulation in ChR2 mice at both frequencies evoked goal-directed escapes

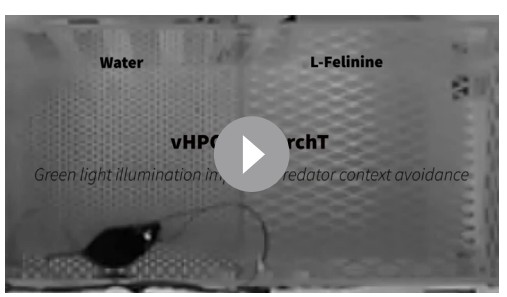

**Video 6.** HPC-AHN pathway inhibition impairs the retrieval of contextual memory of predator cue. HPC-AHN ArchT animals do not avoid the L-felinine paired chamber during the green light illumination. HPC-AHN GFP animals display avoidance of the L-felinine chamber and display predator cue associated chamber.
https://elifesciences.org/articles/74736/figures#video6

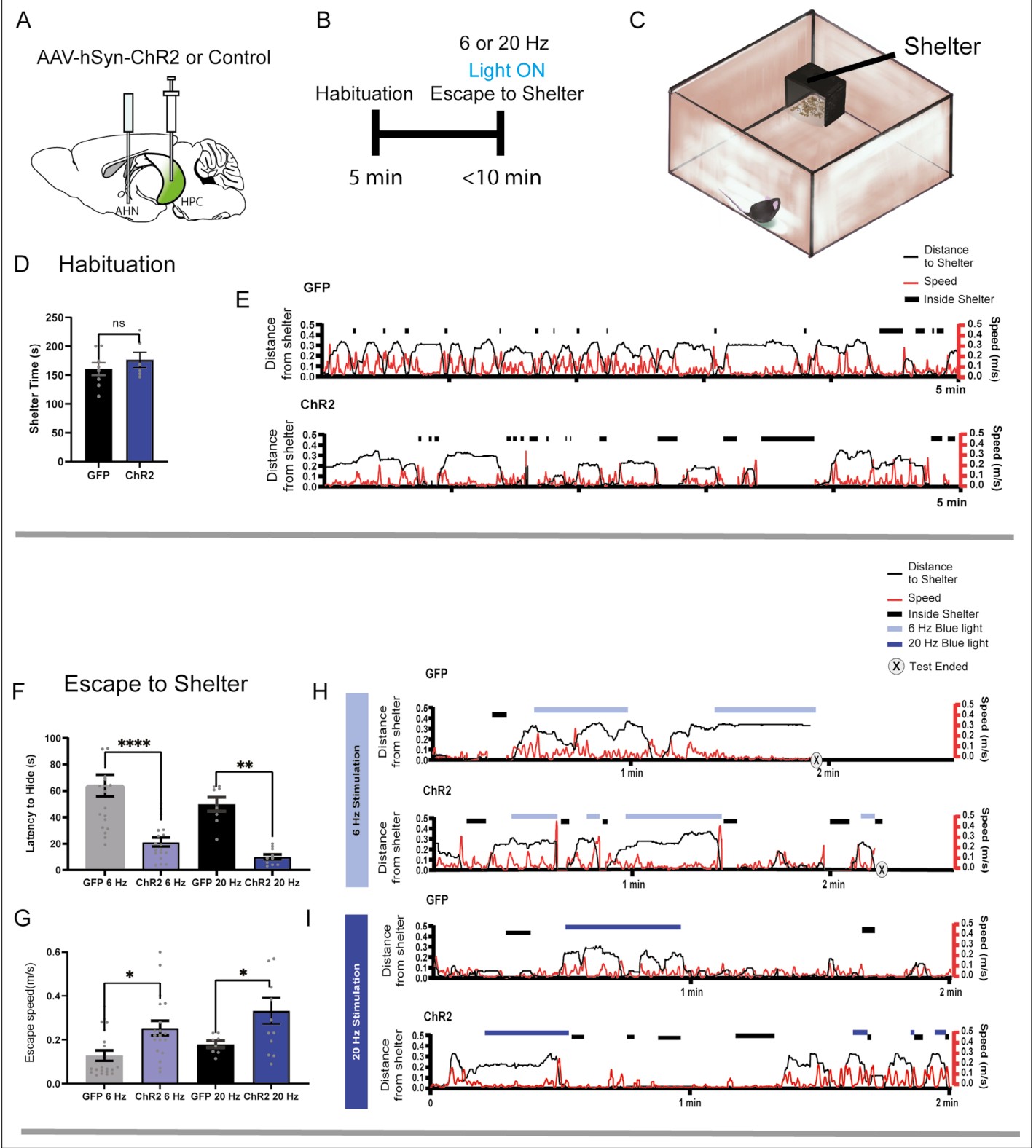

**Figure 7.** HPC→AHN pathway activation induces goal-directed escape. (**a**) Schematic illustration of optogenetic HPC terminal activation in the AHN. (**b**) Schematic describing a test paradigm consisting of habituation (5 min) and a 6 or 20 Hz stimulation stage to induce shelter-directed escapes. (**c**), A cartoon drawing of the open field arena with a shelter box. (**d**) Time spent in the shelter during habituation (GFP N=8, ChR2 N=6, unpaired t-test, two-tailed, t=0.9241, df=12, p=*0.3736*, NS). (**e**) Representative line graphs for GFP (top) and ChR2 (bottom) mice, showing distance from shelter (black lines), speed (red lines), and moments when mice were inside the shelter (black boxes) over the 5 min habituation period. (**f**) Latency to escape to the

*Figure 7 continued on next page*

*Figure 7 continued*

shelter after optogenetic HPC terminal activation. (two-way ANOVA, frequency x genotype, $F_{(1, 52)} = 0.04138$, p=0.8396, NS, frequency effect, $F_{(1, 52)}=3.268$, p=0.0764, NS, genotype effect, $F_{(1, 52)}=34.71$, ****p<0.0001, Sidak's multiple comparisons test, 6 Hz GFP vs. ChR2, ****p<0.0001, 20 Hz GFP vs. ChR2, **p=0.0023). (**g**) Speed of escape running. 2-WAY ANOVA, frequency x genotype, $F_{(1, 55)} = 0.1134$, p=0.7375, NS, frequency effect, $F_{(1, 55)}=2.78$, p=0.1011, NS, genotype effect, $F_{(1, 55)}=12.65$, ***p=0.008, Sidak's multiple comparisons test, 6 Hz GFP vs. ChR2, *p=0.0139, 20 Hz GFP vs. ChR2, *p=0.0413. (**h, i**) Representative line graphs for GFP and ChR2 mice, showing distance from shelter (black lines), speed (red lines), and moments when mice were inside the shelter (black boxes) during the 6 Hz (**h**) and 20 Hz (**i**) HPC terminal stimulation stage. Light and dark blue highlights indicate the duration of 6 Hz and 20 Hz light stimulation, respectively, and (x) denotes test termination time. All results reported are mean ± s.e.m. *p < 0.05, **p < 0.01, ***p < 0.001, ****p<0.0001.

The online version of this article includes the following source data for figure 7:

**Source data 1.** Numerical data shown in *Figure 7*.

toward the shelter, with a shorter latency to escape and a greater speed of escape running compared to GFP controls (*Figure 7f–i*, *Videos 7 and 8*). Thus, our findings show that the behavioral response evoked by HPC→AHN pathway activation is not just a simple increase in locomotion but constitutes a goal-directed escape toward a safe shelter.

## Optogenetic inhibition of HPC→AHN pathway impairs goal-directed escapes to shelter

Like predator odors, highfrequency (17–22 kHz) ultrasound stimuli evoke strong defensive responses in mice, including escape flight and freezing (*Wöhr and Schwarting, 2010*; *Fendt et al., 2018*; *Sales, 2010*). A recent study found that the same ultrasound stimulus elicits different defensive responses depending on the availability of a safe shelter; mice display escape flights when a safe shelter is available but freezing when there is no shelter (*Vale et al., 2017*). The study suggests that animals' spatial knowledge about escape routes and shelter availability determines the best course of defensive actions in the face of predatory threats.

Given the role of animals' spatial knowledge in shaping escape responses, we tested whether the HPC→AHN pathway activity is necessary for mice to use mnemonic information about shelter availability and location during the ultrasound-evoked escape (*Figure 8a and b*). During a habituation stage (7 min), mice were allowed to explore a modified Barnes maze with 20 equally spaced holes, one of which leads to a shelter box (*Figure 8b and c*). Both ArchT and GFP control mice found the shelter at least once during the survey stage and spent a comparable amount of time in the shelter

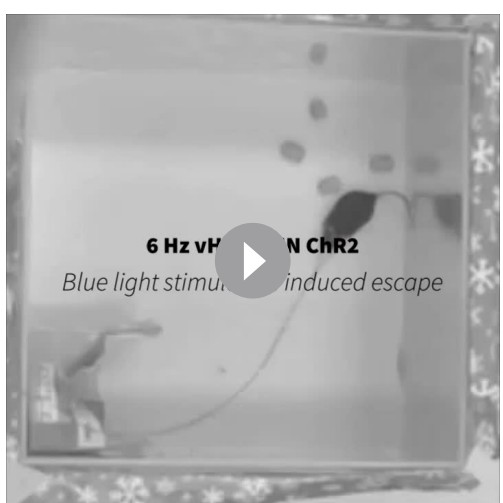

**Video 7.** Low frequency (6 Hz) stimulation of HPC-AHN pathway induces escape to shelter. HPC-AHN ChR2 animals display escape to shelter when 6 Hz photostimulation is delivered. HPC-AHN GFP animals do not escape to shelter upon 6 Hz photostimulation.
https://elifesciences.org/articles/74736/figures#video7

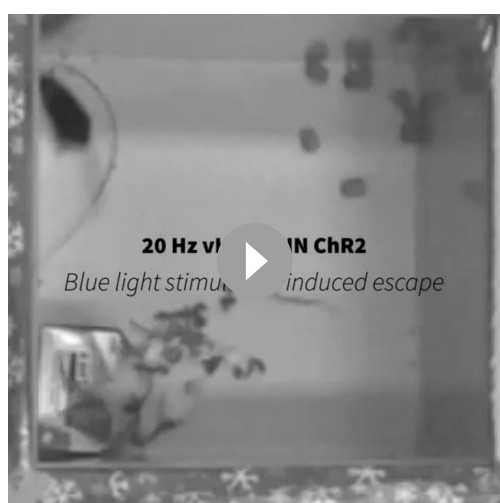

**Video 8.** High frequency (20 Hz) stimulation of HPC-AHN pathway induces escape to shelter. HPC-AHN ChR2 animals display escape to shelter when 20 Hz photostimulation is delivered. HPC-AHN GFP animals do not escape to shelter upon 20 Hz photostimulation.
https://elifesciences.org/articles/74736/figures#video8

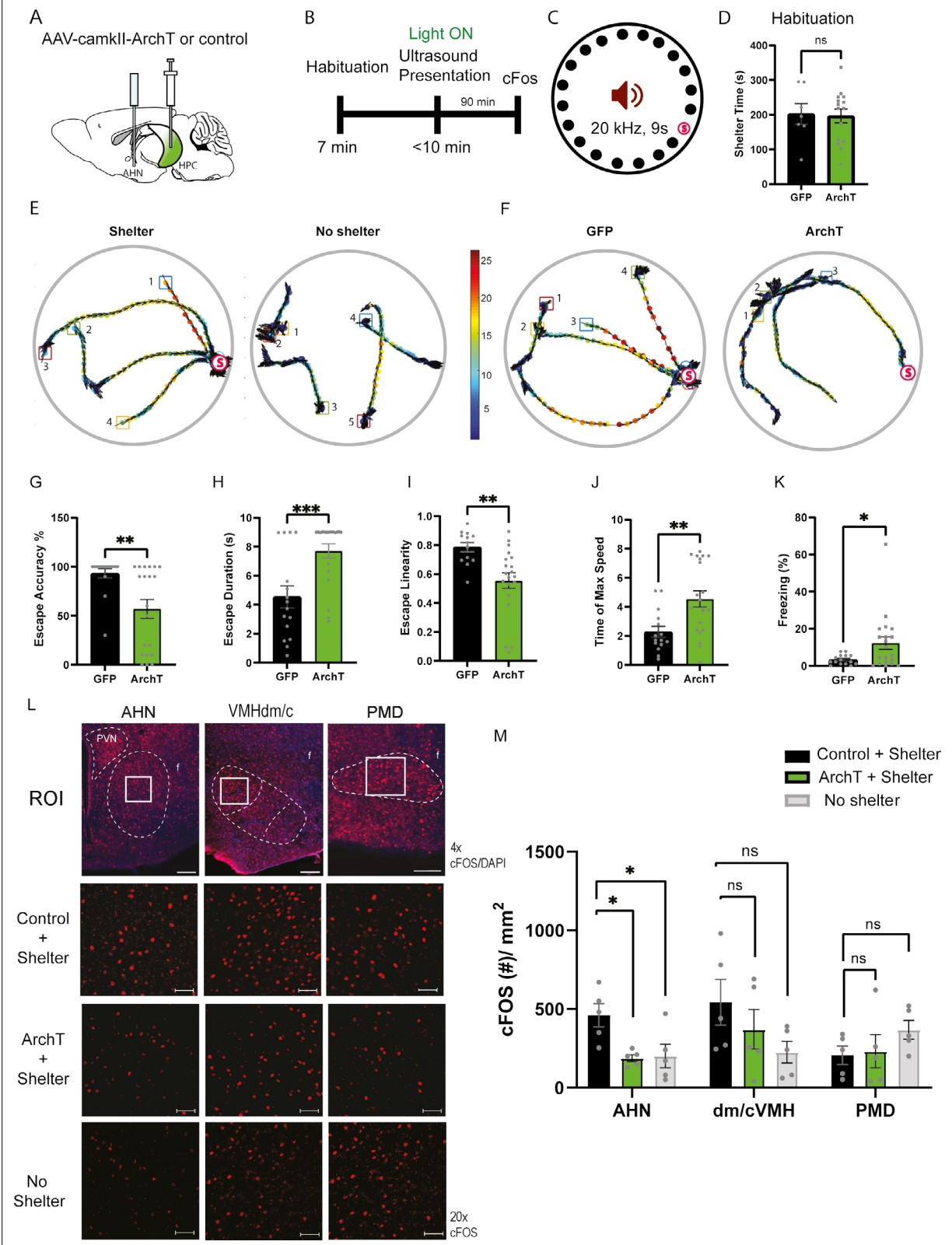

**Figure 8.** HPC→AHN pathway is necessary for goal-directed escape. (**a**) Schematic illustration of optogenetic HPC terminal inhibition in the AHN (GFP N=7 and ArchT N=15). (**b**) Schematic describing a test paradigm consisting of habituation (7 min) and a threat delivery stage during which ultrasound (20 kHz, 9s) is turned on after mice voluntarily come out of the shelter to induce a shelter-directed escape. (**c**), Top view of testing apparatus, a modified Barnes maze. 's' denotes the position where a shelter was placed. The red speaker sign denotes the auditory threat played from a speaker above the

*Figure 8 continued on next page*

*Figure 8 continued*

apparatus centre. (**d**) Time spent in the shelter during habituation stage (unpaired t-test, t=0.1698, df=20, p=0.8668, NS). (**e**) Representative ultrasound-evoked escape trajectories for a Wild type (WT) when a shelter is available (left) vs. WT when no shelter is available (right). (**f**) Representative ultrasound-evoked escape trajectories for a GFP control (left) and ArchT (right). (**e,f**) Individual threat presentation as a trial is numbered next to a square box, which denotes the animals' starting position at the beginning of 9 s of 20 kHz sound. Dot color along the trajectory lines reflects animals' speed. The arrows track animals' head direction. The heatmap colorbar displays the scale of speed (pix/s). (**g**) Accuracy of reaching the shelter during escape. (unpaired t-test, t=3.149, df=33, \*\*p=*0.0034*). (**h**), The linearity of escape trajectories expressed as the percentage ratio between the length of escape trajectory and a linear distance from escape onset position (i.e. ultrasound onset) to the shelter (unpaired t-test, t=3.266, df=31, \*\*p=*0.0027*). (**i**) Time elapsed from the ultrasound onset to the shelter arrival (unpaired t-test, t=3.666, df=33, \*\*\*p=*0.0008*). (**j**) Time elapsed to reach the maximum speed during escape running to the shelter (unpaired t-test, t=3.134, df=31, \*\*p=*0.0036*). (**k**) Time spent in freezing between the ultrasound onset and the shelter arrival (unpaired t-test, t=2.261, df=33, \*p=*0.0305*). (**l**) c-Fos immunochemical detection across the medial hypothalamic defense system (AHN, VMHdm/c, PMD). First row, representative 4x epi-fluorescence microscope images of the medial hypothalamic defense system in GFP control mice. The regions of interest (ROI, white squares) within AHN, VMHdm/c, and PMD were imaged by confocal microscopy for cell counting. Second and third row: representative 20x confocal images of c-Fos signals in AHN, VMHdm/c, and PMD activated by ultrasound-evoked escapes with a shelter available in GFP (second row) and ArchT mice (third row). Fourth row: representative 20x confocal images of c-Fos signals activated by ultrasound-evoked escapes without shelter in controls. (**m**) Density of c-Fos signals activated by ultrasound-evoked escapes in AHN, VMHdm/c, and PMD in GFP controls with shelter (black), ArchT mice with shelter (green), and controls without shelter (grey) (N=5 mice for each group). AHN (1-WAY ANOVA, F(2,12)=6.171, \*p=*0.0144*, Sidak's multiple comparison test, Control Shelter vs. ArchT Shelter, \*p=*0.0177*, Control Shelter vs. Control No Shelter, \*p=*0.0237*), VMHdm/c (one-way ANOVA, F(2,12)=1.824, p=0.2035, NS, Sidak's multiple comparison test, Control Shelter vs. ArchT Shelter, p=0.5391, NS, Control Shelter vs. Control No Shelter, p=0.1548), PMD (one-way ANOVA, F(2,12)=1.25, p=0.3212, NS, Sidak's multiple comparison test, Control Shelter vs. ArchT Shelter, p=0.9665, NS, Control Shelter vs. Control No Shelter, p=0.3058, NS). Two-way ANOVA (ROI x treatment, F(4,36)=2.347, p=0.0729, NS, ROI effect, F(2,36) = 1.391, p=0.262, NS, treatment effect, F(2,36)=2.44, p=0.1015, NS, Sidak's multiple comparisons test). AHN (Control Shelter vs. ArchT Shelter, p=0.1064, NS, Control Shelter vs. No Shelter, p=0.1344, ArchT Shelter vs. No Shelter, p=0.9993, NS), VMHdmd/c (Control Shelter vs. ArchT Shelter, p=0.4470, NS, Control Shelter vs. No Shelter, \*p=*0.0476*, ArchT Shelter vs. No Shelter, p=0.5876, NS), PMD (Control Shelter vs. ArchT Shelter, p=0.9957, NS, Control Shelter vs. No Shelter, p=0.4999, NS, ArchT Shelter vs. No Shelter, p=0.6378, NS). Scale bar = 100 µm for 4x epi-fluorescence microscope images and 10 µm for 20 x confocal images. (PVN, paraventricular nucleus. f, fornix).

The online version of this article includes the following source data and figure supplement(s) for figure 8:

**Figure supplement 1.** Mice display shelter-directed escape or freezing depending on the shelter availability.

**Figure supplement 1—source data 1.** Numerical data shown in *Figure 8—figure supplement 1*.

**Figure supplement 2.** HPC→AHN pathway inhibition impairs ultrasound (US)-evoked escape responses.

**Figure supplement 2—source data 1.** Numerical data shown in *Figure 8—figure supplement 2*.

**Figure supplement 3.** HPC→AHN pathway inhibition does not change anxiety-related behaviors.

**Figure supplement 3—source data 1.** Numerical data shown in *Figure 8—figure supplement 3*.

---

(*Figure 8d*), indicating that the two groups had a similar condition to memorize shelter location and shelter availability. During the subsequent threat delivery stage, light illumination (i.e., pathway inhibition) started right after mice voluntarily came out of the shelter, and a 9 s ultrasound stimulus (20 kHz) was triggered manually at randomized positions on the platform. Upon hearing the ultrasound threat, mice first turned its heads towards the shelter and initiated an escape flight, reaching a maximum running speed at the middle of the escape path, which are the known features of the goal-directed escape flight (*Figure 8e*, shelter, *Figure 8—figure supplement 1a-c*). Consistent with a previous report, mice displayed these characteristic escape responses only when a shelter is available. When the shelter was removed from the maze before the survey stage, the ultrasound threat failed to elicit escape flights, but instead caused either freezing or slow and disorganized flights in random directions (*Figure 8e*, no shelter, *Figure 8—figure supplement 1d-h*).

Optogenetic inhibition of the HPC→AHN pathway produced a range of effects on the goal-directed escape. Instead of a quick and direct flight to shelter seen in GFP controls, ArchT mice displayed disorganized escape trajectories and slow escape running speed, reminiscent of how the control mice respond to ultrasound when the shelter is not available (*Figure 8f*, *Figure 8—figure supplement 2a*,b, *Video 9*). In addition, ArchT mice directed their flights to locations farther away from the target shelter (i.e. lower escape accuracy, *Figure 8g*), resulting in low escape success rate of 30% (6 out of 20 trials) compared to 87% in GFP controls (13 out of 15 trials) (*Figure 8—figure supplement 2c*,d). The lower escape accuracy and rate of successful escape suggest that ArchT mice failed to use a memory of shelter availability and shelter location to support their goal-directed escapes. Furthermore, a decrease in the organization and efficiency of escape was indicated by changes in various parameters such as escape linearity, escape duration, time to reach the maximum speed and increased freezing

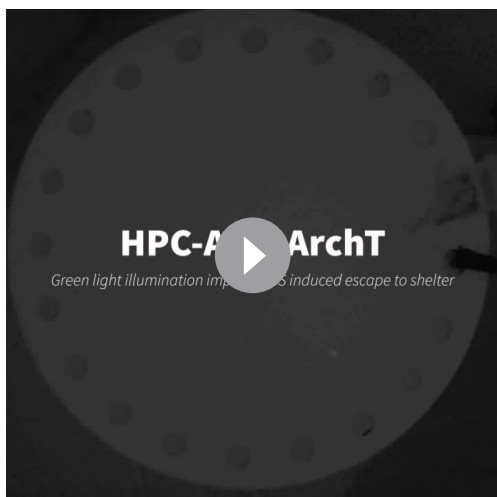

**Video 9.** HPC-AHN pathway inhibition impairs goal-directed escape to shelter. HPC-AHN ArchT animals display fragmented and impaired escape to shelter compared to the GFP controls upon hearing the 20 kHz ultrasound during the green light illumination. https://elifesciences.org/articles/74736/figures#video9

(*Figure 8h–k*). The impairments in goal-directed escape were not accompanied with any changes in anxiety-related behaviors (*Figure 8—figure supplement 3*).

Lastly, we investigated the role of HPC inputs in driving AHN activities during the goal-directed escape. ArchT mice were exposed to an ultrasound-evoked escape flight coupled with the HPC→AHN pathway inhibition (ArchT/Shelter+). Control mice were split into two groups, where one group was exposed to an ultrasound-evoked escape flight with a shelter available (Control/Shelter+) and the other group without shelter (Control/ Shelter-). Nienty min later, mice were euthanized for immunochemical detection of c-Fos. We found that the inhibition of HPC→AHN pathway and removal of shelter significantly reduced in the c-Fos measure in the AHN compared to the control condition (Control/Shelter+), whereas no change was detectable at the level of VMHdm/c and PMD (*Figure 8l and m*). This suggests that the AHN is not activated properly during the ultrasound-evoked escape if the HPC→AHN pathway is inhibited, or if a shelter is not available. Thus, the inhibition of HPC→AHN pathway and the removal of shelter during the ultrasound-evoked escape produced similar effects not only at the behavioral level but also on the AHN activity. Taken together, these results demonstrate that the HPC→AHN pathway supports the goal-directed escape by driving the AHN activity with mnemonic information about shelter availability and shelter location.

## Discussion

Current knowledge of HPC control of fear and defensive response has largely been derived from studies of associative memory for nociceptive stimuli (e.g. electric foot shocks) (*Siegel and Pott, 1988*; *Sales, 2010*; *Maren and Quirk, 2004*). While informative, they have left a widening gap between our understanding of the neural circuit mechanism underlying fear and the complex innate defensive behaviors displayed in natural environments. Our investigation of the HPC-AHN pathway provides a framework on how explicit memory and transmission of contextual information control innate defensive behaviors. To our knowledge, the present study is the first to (1) delineate a direct functional connection between the hippocampus and the medial hypothalamic defense system and to (2) show how hippocampal signals representing environmental contexts control innate defensive responses at the neural circuit level.

We found that a direct optogenetic stimulation of the entire AHN structure increases avoidance, immobility, and escape running and jumping. This is consistent with a recent finding that the selective activation of excitatory VMHdm/c inputs to the AHN elicits escape running, immobility, and jumping (*Wang et al., 2015*). Interestingly, however, the selective activation of VMHdm/c inputs to the dorso-lateral periaqueductal gray produced only immobility but failed to evoke escape running and jumping. These findings, together with ours, support the idea that distinct aspects of defensive responses to threat are mediated by different cell types and efferent projections in the VMHdm/c and AHN. The AHN has been shown to be a largely GABAergic structure, with some scattered glutamatergic cells located at the ventral aspect of the medial zone (*Anagnostaras et al., 2001*; *Boudaba et al., 1996*). Thus, it remains to be investigated whether VMHdm/c inputs selectively target AHN GABA or glutamatergic cells, or both. Of note, we did not observe any escape jumping upon stimulating AHN GABA cells (data not shown). We speculate that the activity of AHN glutamatergic, but not GABAergic, cells may be sufficient to evoke escape jumping, and that the escape jumping induced by the VMHdm/c-AHN pathway may have been driven at least in part by inputs to AHN glutamatergic cells.

Several tract-tracing studies, including ours, have shown that the medial hypothalamic defense system receives strong excitatory inputs from the hippocampus (*Takahashi et al., 2005*; *Kishi et al., 2000*). Our anterograde tracing experiments revealed that hippocampal axon terminals are found almost exclusively at the AHN, but not the PMD and VMHdm/c. This suggests that the AHN is the main entry site for hippocampal signals in the hypothalamic defense system, therefore an ideal brain region for integrating environmental context and concomitant predator sensory information to support the contextual memory of predator threats. Consistently, a previous work found that predatory context (e.g. cat-associated context) induces a robust increase in c-Fos level in the AHN (*Cezario et al., 2008*). Interestingly, the same study identified the PMD, not the AHN, as the most responsive hypothalamic region to predatory context despite the relative scarcity of hippocampal innervation in the PMD. Furthermore, a recent study by *Wang et al., 2021a* found that the PMD activation evokes an organized escape in which mice quickly assess environment's layout and find efficient flight path, whereas the activation of other hypothalamic nuclei, including the AHN and VMHdm/c, only induced stereotyped panic-related escape responses such as running and jumping (*Wang et al., 2021b*). We found that a direct AHN activation evokes panic-related running and jumping in an empty open field arena with no shelter. However, once a shelter was added and mice formed a memory of shelter availability during the habituation stage, the same AHN activation evoked an organized goal-directed escape to shelter, instead of jumping and running. This seemingly contradicting result between Wang et al and our study may be due to a difference in study design; in Wang et al, mice naive to the environment's layout received AHN stimulations before they fully formed the memory of surrounding, whereas in our study mice received AHN stimulations after the shelter memory was encoded. It is still possible that the AHN and the PMD act together to support the contextual memory of predator threat, where the AHN first receives contextual information from the hippocampus and then conveys it to the PMD. This possibility will have to be tested by selectively blocking AHN inputs to the PMD and analyzing its impact on the contextual memory of predatory threats.

Electrophysiological recordings confirmed the monosynaptic nature of the HPC→AHN connection. In slice recordings experiments, patching was guided by mCherry fluorescence in double transgenic (Dlx5/6-Flpe; Frepe) reporter mice. The Dlx5/6-Flpe line has been used to label GABA cells in the forebrain cortex and hippocampus with a high labeling efficacy and specificity (*Nguyen et al., 2020*; *Taniguchi et al., 2011*; *Whissell et al., 2019*; *Dedic et al., 2018*; *Esteban Masferrer et al., 2020*). Optogenetic stimulation of hippocampal axon fibers in the AHN evoked robust EPSPs in both GABAergic and non-GABAergic cells with an onset latency less than 5ms that indicates monosynaptic responses. It should be noted, however, the labeling efficacy of the Dlx5/6-Flpe mouse line has not been thoroughly characterized in the hypothalamic regions, including the AHN. Thus, our study may overestimate the abundance of non-GABAergic AHN cells receiving monosynaptic hippocampal inputs. Experiments using other reporter strains such as GAD67-GFP mice will help to further clarify the abundance of hippocampal inputs to non-GABAergic cells in the AHN.

Compared to a direct AHN stimulation which invariably induced escape jumping, HPC→AHN pathway activation only increased running bouts and speed. This suggests that more intense escape responses such as jumping likely require additional inputs to the AHN from other structures such as amygdala and VMHdm/c that encode sensory information about predatory threats. Despite not being strongly innervated by the hippocampus, the VMHdm/c receives direct inputs from the medial and basolateral amygdala areas that process multimodal sensory information about predatory threat. Consistently, single-unit recordings showed that the VMHdm/c is activated earlier as mice approach a predatory threat, suggesting that its firing rate likely encode sensory aspects of threat intensity and threat distance (*Esteban Masferrer et al., 2020*). Importantly, the activation of HPC→AHN pathway was as powerful as direct AHN stimulation in producing a strong real-time and conditioned place aversion, suggesting that the pathway can form a lasting long-term memory of threat-associated environmental context. Indeed, we found that upon the inhibition of HPC→AHN pathway, mice failed to remember where a predator cue was previously encountered. This indicates that the hippocampus controls the medial hypothalamic defense system and mediates the contextual memory of predator threats via its direct projections to the AHN. Aversive experience such as the predator odor can induce remapping of place cell firing which becomes stabilized after 24 hr. Thus, HPC→AHN pathway is likely activated by a specific hippocampal place cell ensemble that represents predator odor location and context. It is of note, however, that our optogenetic inhibition targeted only the retrieval phase of

memory in the conditioned place aversion. Thus, it remains unknown if the HPC→AHN pathway is also involved in the memory encoding and how the synaptic plasticity of the HPC inputs to AHN changes during memory encoding and/or consolidation.

HPC→AHN pathway activation evoked goal-directed escapes, whereas its inhibition disrupted a successful escape to shelter. In the ultrasound-evoked escape assay, mice detect an ultrasound threat and then evaluate whether shelter is available based on their memory of shelter. If shelter is available, mice compute the flight direction before launching an escape, and if not, they freeze. Importantly, a previous study using the same assay has shown that unlike in Morris water maze or Barnes maze tests, mice can make accurate escape flights even in complete darkness, suggesting that external landmarks (reference memory) are not required for mice to determine the shelter location. Instead, mice compute an escape vector to the shelter location by integrating self-motion over time using the path integration strategy (*Vale et al., 2017*; *Vale et al., 2020*; *Bang et al., 2012*). We found that instead of making a quick and direct escape to shelter, ArchT mice display a slow escape running and are more likely to flee to unsafe locations away from the target shelter. The observed behavioral impairments present multiple possibilities regarding the role of HPC→AHN pathway. The most plausible explanation is that the pathway encodes a short-term memory of the shelter availability to increase a motivational drive to escape. When mice hear the ultrasound threat, the hippocampus may reactivate shelter memory and send the signals to the AHN, thereby increasing the escape drive and escape-associated locomotion. If the pathway is optogenetically inhibited, however, the shelter memory recall would no longer be able to activate the AHN and support goal-directed locomotion, resulting in a slow or even lack of escape running. Alternatively, the HPC→AHN pathway activity may be necessary for spatial navigation during a goal-directed escape by encoding specific geometric information about a shelter location generated by the hippocampus. Such information may be used for the path integration process either within the medial hypothalamic defense system or its downstream targets such as the dorsal periaqueductal gray to compute an escape vector. A recent study found that the retrosplenial cortex (RSP) input to superior colliculus (SC) plays an important role in shelter-directed escape by continuously encoding egocentric representation of shelter direction (*Vale et al., 2020*). It remains to be tested whether the medial hypothalamic defense system and the RSP-SC pathway project to the same postsynaptic cells in the dorsal periaqueductal gray to support an organized escape to shelter. It is plausible that the HPC-AHN pathway controls the motivational drive to escape based on shelter availability while the RSP-SC pathway controls the escape direction based on shelter location.

## Materials and methods

**Key resources table**

| Reagent type (species) or resource | Designation | Source or reference | Identifiers | Additional information |
|---|---|---|---|---|
| Strain, strain background (*Mus musculus*, male) | AHN-ChR2/GFP; HPC-AHN ChR2/GFP; HPC-AHN ArchT/GFP | Charles River | C57BL/6 | |
| Strain, strain background (*Mus musculus*) | GABA-mCherry or Dlx5/6-FLPe;RC::FrePe | PMID:22151329 | JAX#029486 x JAX#010815 | Obtained by crossing homozygous RC::FrePe6 x Dlx5/6-FLPe mice |
| Recombinant DNA reagent | AAV2/9-hSyn-ChR2-eYFP (ChR2) | Addgene Vrial Vector Core | 26,973 | |
| Recombinant DNA reagent | AAV2/9 or AAV2/5-CB7-CI-eGFP (Control) | Addgene Vrial Vector Core | 105,542 | |
| Recombinant DNA reagent | AAV2/5-camkIIa-ArchT-GFP (ArchT) | Addgene Vrial Vector Core | 99,039 | |
| Antibody | anti-GFP (chicken polyclonal) | Abcam | ab13970 | (1:1000) |
| Antibody | anti-cFOS (rabbit polyclonal) | Santa Cruz Biotechnology | SC-52 | (1:1000) |
| Antibody | Alexa Fluor 594-conjugated anti-rabbit secondary antibody (donkey polyclonal) | Jackson ImmunoResearch Laboratories | AB_2340621 | (1:500) |

*Continued on next page*

*Continued*

| Reagent type (species) or resource | Designation | Source or reference | Identifiers | Additional information |
|---|---|---|---|---|
| Antibody | Alexa Fluor 488-conjugated anti-chicken secondary antibody (donkey polyclonal) | Jackson ImmunoResearch Laboratories | AB_2340375 | (1:1000) |
| Chemical compound, drug | DAPI | Cell Signaling Technology | 4,083 S | |
| Chemical compound, drug | L-Felinine | Toronto Research Chemicals | F231250 | |
| Software, algorithm | DeepLabCut | PMID:31227823 | | |
| Software, algorithm | ANY-MAZE | StoeltingCo | | |
| Software, algorithm | MATLAB | Mathworks | | |
| Software, algorithm | Prism | GraphPad | | |
| Software, algorithm | Code for MATLAB | Custom written code | 10.5281/zenodo.5899428 | Custom written code for MATLAB used for ultrasound evoked escape |

## Animals

Adult C57BL/6 male mice (Charles river) at 8–12 weeks of age were used for AHN soma activation (GFP N = 7, ChR2 N = 6), HPC terminal activation or inhibition studies (GFP N = 21, ChR2 N = 22; GFP N = 15, ArchT N = 22) and optogenetic electrophysiological confirmation study (ChR2 N = 3). Double transgenic Dlx5/6-FLPe;RC::FrePe male (N = 3) and female (N = 2) mice were obtained for electrophysiology experiment by crossing homozygous RC::FrePe (*Bang et al., 2012*) mice with Dlx5/6-FLPe mice [Tg(mI56i- FLPe)39Fsh/J, JAX#010815]. The RC::FrePe is a dual-recombinase responsive fluorescent allele containing a frt-flanked STOP and loxP-flanked mCherry::STOP that prevent transcription of GFP. FLP recombinase results in mCherry expression, and further exposure to Cre recombinase results in GFP expression in the overlapping cell populations that express both Cre and FLP.10 days prior to testing, animals were single housed with food and water provided ad libitum in 12 hr light/dark cycle. All procedures were approved by the Local Animal Care Committee (LACC, AUP#20011332) at University of Toronto.

## Viral vectors and stereotaxic surgery

AAV2/9-hsyn-hChR2 (H143R)-eYFP, AAV2/5-camk2a-eArchT3.0-GFP, AAV2/9-CB7-CI-eGFP (GFP control), and AAV2/5-CB7-CI-eGFP (GFP control) were purchased from the Addgene Viral Vector Core and used as received. For all surgical procedures, mice were anesthetized with isoflurane (4% for induction and 2% for maintenance of anesthesia) at an oxygen flow rate of 1 L/min, and head fixed in a stereotactic frame (David Kopf). Eyes were lubricated with an ophthalmic ointment throughout the surgeries. Ketoprofen was provided for pain management during post-operative recovery. Viruses were infused by pressure injection. For the AHN infusion (AP –0.85 mm, ML 0.45 mm, DV –5.2 mm), 69 nL per site was infused by a pulled glass needle and Nanoject II (Drummond Scientific) at 46 nl/s rate, and the needle was left in place for additional 10 min to limit the virus drag during needle retract. For the ventral hippocampus/subiculum infusion (AP –3.8 mm, ML –2.1 mm, DV –4.8 mm, 10° away from the midline), 300 nL per site were infused by cannula needle connected to Tygon tubing to a 10 µL Hamilton syringe (Hamilton Company) at rate 0.1 µl/min. Custom made ferrule fibers consisting of optic fibers (200 µm core diameter, 0.39 NA, Thorlabs) threaded in 1.25 mm wide zirconia ferrules (Thorlabs) were implanted at the AHN (AP –0.85 mm, ML 1.38 mm, DV –5.1 mm, 10° towards the midline) 2 weeks after the viral infusion surgery. All animals were handled for a minimum of 5 minutes for 3 days prior to behavioral testing 2 weeks post implant operation.

## Optogenetic manipulation

For bilateral light delivery, a patch cable (200 µm core diameter, 0.37 NA; Doric Lenses) was connected to a 1 × 2 optical commutator (Doric Lenses) to divide the light path into two arena patch cables attached to the implanted optic fibers. For ChR2-mediated optogenetic stimulation, blue light

(473 nm, 6 Hz or 20 Hz) was produced using an arbitrary waveform generator (Agilent, 33,220 A) and a diode-pumped solid-state laser (Laserglow) at a power intensity of 5 mW from the optic fiber tip. The same animals were used among the different tasks (escape in open field, real-time place aversion, conditioned place aversion, goal-directed escape to shelter) and frequencies (6 or 20 hz). 6 or 20 Hz frequency optostimulations were delivered in a counterbalanced manner for both GFP and ChR2 animals for each behavioral paradigm with minimum 3 days of inter-test interval. For ArchT-mediated optogenetic inhibition, green light (532 nm, Laserglow) was applied continuously at a power intensity of 15 mW from the optic fiber tip. Light power was measured at the optic fibre tip using a power meter (PM121D, Thorlab) before each behavioral test.

## Optostimulation-evoked escape responses in open field and physical restraint

Following a 5-min habituation to the tethering cable in home cage, animals were placed in a clear plexiglass chamber (short walled- escapable condition: 50cm x 50 cm x 20 cm) or in an opaque walled chamber (inescapable condition: 30cm x 30 cmx 30 cm). In the clear chamber, low- and high-frequency photostimulation effects were compared while keeping consistent light power (5 mW) at the optic fibre tip. Animals were given two 2-min photostimulation (6 Hz, 5ms pulse width) each followed by 2 min off period to observe the light offset effect. Rearing, jumping, freezing, and grooming were blindly and manually scored by key press in ANY-MAZE (Stoelting Co) for all animals. Speed increase was calculated as a normalized difference in speed from light on and off: [(Speed during light ON-Speed during OFF)/(Speed during light ON + Speed during OFF)]. Two weeks after testing, the effects of AHN stimulation in the chambers (condition easy and hard), animals were tested in a physical restraint. DecapiCones (Braintree Scientific) were cut around head and shaved neck to create spaces for arena cable linked to animals' head cap and for collar sensor with a pulse oximeter (STARR Life Sciences). The collar sensor collected movement as a binary value (0 = no movement, 1 = movement). The struggle index was calculated as the movement values taken from 10 s bins of the total of physical restraint. Animals were restrained in the DecapiCones during physical restraint and received AHN stimulation (473 nm blue light, 20 Hz, 10 s ON, 10 s OFF) for 30 min.

## Real-time place aversion (RTPA) and conditioned place aversion (CPA)

A custom-made 45 cm × 20 cm × 35 cm apparatus was equally divided such that each side possessed a distinct visual context. After 5 min of habituation, the preferred chamber was selected as the stimulation chamber. Animals received either 6 or 20 Hz blue light illumination upon entering the stimulation chamber during a 20-min RTPA test. Twenty-four hr after the RTPA test, animals were re-introduced to the two-chamber apparatus with light off, and the preference during the first 5 min was analyzed to measure the retrieval of CPA memory. Animals were placed back into the home cage for 5 min and reintroduced to the testing apparatus to begin a second RTPA and CPA with 6 or 20 Hz stimulation. The light was paired to the opposite side of the prior RTPA session. The order of stimulation frequency was pseudorandomized. ANY-MAZE software was used to determine the amount of time spent in each chamber and their corresponding track plots. The calculations for the RTPA and CPA: Place preference indexes for pre-conditioning = [(Time spent in preferred side - Time spent in less preferred side)/Total chamber time]. Place aversion index for post-conditioning = [(Time spent in stimulation side - Time spent in no stimulation side)/Total chamber time]. Conditioned place aversion index = Aversion index of Post-Conditioning – Aversion index of Pre-Conditioning. Conditioned place aversion assessed the change in place preference index before and after the conditioning session.

## Predator odor contextual fear conditioning

A custom-made 45 cm × 20 cm × 35 cm apparatus was equally divided such that each side possessed a distinct visual context. Two different testing paradigms (P1 and P2) were employed. Mice were handled (5 min for 3–5 days) before both paradigms and habituated to the testing room for 3 days. Both tests consisted of 'pre-conditioning' phase (5 min), a habituation period to the two-chamber apparatus. The preferred chamber was always paired with L-Felinine (F231250, Toronto Research Chemicals). Freshly prepared L-Felinine 10% in water (40 µl) or water (40 µl) was pipetted on a filter paper of a petri dish. Animals were agitated and defensive toward the experimenter after each L-Felinine pairing, thus cup/tunnel handling was used to minimize stressful handling experience. All

behavioral and tracking analysis were done using ANY-MAZE software. Manual behavioral scoring was done blindly to the treatment post experiment by changing the file names and randomizing video sequence. Freezing was quantified as no locomotive movement besides respiration. Two paradigms were used to increase replication of the predator odor-context memory fear impairment. In the P1 (AM/PM conditioning design), mice were enclosed in either L-Felinine (10 %)- or water-paired chamber for odor-context pairing for 20 minutes in AM and PM in a counterbalanced manner for two days (Xia et al., 2017). On day 4, animals were placed back in the two-chamber apparatus and measured for defensive behaviors and chamber preference. The P1 allowed testing for predator odor contextual fear memory and the paradigm 2 (Figure 6—figure supplement 5) additionally measured place aversion development after each predator odor-context pairing. A total of 6 predator odor-context pairings were carried out from day 0. Each day consisted of 5 min of free exploration, followed by a 6-min L-Felinine (0.3%) pairing to the context of preferred chamber side. Aversion index was calculated as: Aversion index = [(Time spent in preferred side - Time spent in less preferred side)/Total chamber time]. Conditioned place aversion assessed the change in place aversion index before and after the L-Felinine conditioning sessions as: Conditioned place aversion index = Aversion index of Post-Conditioning – Aversion index of Pre-Conditioning. In AM/PM design (P1), escape running was defined as an event in which mice left the L-Felinine-paired chamber with a peak locomotive speed greater than 50% of the average ambulation speed.

## Optostimulation-evoked goal-directed escape

Mice were introduced to a chamber (40 cm x 40 cm x40 cm) under a dim red light condition (10 lux). A shelter box (12 cm x 12 cm x 8 cm) was placed at a corner of the chamber with home cage bedding material placed inside the shelter box as an olfactory cue. After 5 min of habituation session, a 6 or 20 Hz photostimulation was delivered when the mouse body centre was a minimum 25 cm away from the shelter and the head was not pointing towards the shelter. Mice were tracked with ANY-MAZE software. Tracking error resulting from limited visibility inside a shelter was manually omitted by checking the video. Latency to escape was measured as the time (s) elapsed from the light onset until mouse directed its head and started to move toward the shelter. Speed of escape was measured as the peak speed during escape flight.

## Ultrasound-evoked escape assay

The ultrasound-evoked escape assay, modified from Vale et al., 2017, was conducted under a dim red light condition (10 lux). The behavioral apparatus was a Barnes maze - a white plastic circular platform (92 cm in diameter) with 20 equally spaced holes (5 cm in diameter and 5 cm away from the border of platform) that are blocked by plastic covers. A plastic shelter box (9 cm x 12 cm x 9 cm) was placed at one of the 20 holes with home cage bedding material inside to serve as an olfactory cue. Animals were given a minimum 7 min for the habituation stage, but if they did not find the shelter, they were given an additional 5 min. The ultrasound stimulus (20 kHz sine waveform, 9 s duration, 75 dB) was generated by an amplifier (Topaz AM10) and an ultrasound speaker (L60, Pettersson) positioned 50 cm above the arena. Overhead videos were obtained using a webcam and analyzed using the DeepLabCut to track animals' body parts (nose, centre, and tail base). Locomotion speed, head direction angle (0 ~ 180 degree), and distance between shelter and body parts were calculated with custom-written Matlab scripts. A successful arrival at the shelter was counted when animal's body centre was inside the shelter. Escape accuracy was calculated from how much the shelter target was missed (i.e. how far the animal body was away from the shelter at the end of the 9-s ultrasound stimulus) using an equation [Accuracy = 100%–10% * (distance between body centre and shelter/shelter diameter)]. Freezing behaviors were manually scored while the 9 s of ultrasound was presented before shelter arrival in a treatment blind manner.

## Electrophysiology

Brains were rapidly removed after decapitation and placed into a cutting solution containing the following (in mM): 87 NaCl, 2.5 KCl, 25 NaHCO$_3$, 0.5 CaCl$_2$, 7 MgCl$_2$, 1.25 NaH$_2$PO$_4$, 25 glucose and 75 sucrose (Osmolarity: 315–320 mOsm), saturated with 95% O$_2$/5% CO$_2$. Coronal sections (250 µm thick) containing the hypothalamus were cut using a vibratome (VT-1200, Leica Biosystems). The aCSF solution consisted of the following (in mM): 123 NaCl, 2.5 KCl, 1.25 NaH$_2$PO$_4$, 26 NaHCO$_3$, 10 glucose,

2.5 CaCl$_2$ and 1.5 MgCl$_2$, saturated with 95% O$_2$/5% CO$_2$, pH 7.4, osmolarity 300 mOsm. Slices were recovered at 34 °C in artificial cerebrospinal fluid (aCSF) for 30 min and subsequently kept at room temperature. During experimentation slices were perfused at a rate of 2 ml/min in aCSF and maintained at 27°C–30°C.

Borosilicate glass micropipettes (BF120-69-15, Sutter Instruments) were pulled in a Flaming/Brown Micropipette Puller (P-1000, Sutter Instruments) and filled with an intracellular fluid containing the following (in mM): 108 K-gluconate, 2 MgCl$_2$, 8 Na-gluconate, 1 K$_2$-ethylene glycol-bis(β-aminoethyl ether)-N,N,N',N' -tetraacetic acid (EGTA), 4 K$_2$-ATP, 0.3 Na$_3$-GTP, 10 HEPES (osmolarity: 283–289 mOsm and pH: 7.2–7.4). The resistance of the pipettes was between 3 and 5 MΩ.

Inhibitory post-synaptic currents (IPSCs), and excitatory post-synaptic currents (EPSCs) were blocked using bath application of 100 µM picrotoxin and 10 µM DNQX, respectively. Action-potential-dependent synaptic activity was blocked using 1 µM TTX and monosynaptic release was recovered by subsequent application of 100 µM 4-AP. All recordings were performed on minimum of five animals per group. EPSCs were recorded in voltage-clamp mode with the membrane voltage held at –70 mV. For cell-attached recordings, light stimulation was performed in 5ms pulses of 473 nm blue light at 15 Hz. Light evoked excitatory post-synaptic potentials (eEPSCs) were obtained with a 5ms pulses of 473 nm light with inter stimulus interval at a rate of 2 Hz.

Whole cell patch clamp recordings were obtained using a Multiclamp 700B amplifier (Molecular Devices, California, USA), low pas filtered at 1 kHz and digitized at a sampling rate of 20 kHz using Digidata 1,440 A (Molecular Devices). Data was recorded on a PC using pClamp 10.6 (Molecular Devices) and analyzed using Clampfit (Molecular Devices).

## Histology

For c-Fos immunohistochemistry, mice were anesthetized with avertin 90 min after an exposure to predator context retrieval or ultrasound-evoked escape and underwent transcardial perfusion with 0.1 M phosphate- buffered saline (PBS, pH 7.4), followed by 4% paraformaldehyde (PFA). The brain tissues were removed and were immersed in 4% PFA overnight and cryoprotected in a 30% sucrose solution for 48 hr. Free-floating coronal sections (40 µm) were cut with a cryostat (Leica, Germany), permeabilized with PBS containing 0.3% Triton X-100 (PBS-T) and blocked with 5% normal donkey serum (Jackson ImmunoResearch). The tissues sections were then incubated with primary antibody (rabbit anti-c-Fos, SC-52; Santa Cruz Biotechnology, 1:1000 in PBS-T) at 4 °C for 72 hr. Next, the sections were rinsed with PBS-T and incubated with PBS-T containing Alexa Fluor 594-conjugated donkey anti-rabbit secondary antibody (1:500, Jackson ImmunoResearch Laboratories) at room temperature for 2 hr. Sections were then rinsed with PBS, mounted on slides, and stained with DAPI solution (1 µg/ml in PBS) before coverslipping. Confocal microscope z-stack images were captured using a 20 x objective lens on a LSM800 microscope (Zeiss, Germany). For c-Fos counting, every third section from each animal was captured for each brain region of interest. For AHN (AP –0.70 mm ~ –1.34 mm, total 5 sections per animal), central and ventral regions of AHN below the optic fiber implants were imaged and quantified. For VMH (AP –1.34 mm ~ –1.7 mm, total 4 sections per animal), only the dorsomedial and central areas (VMHdm/c) were included for quantification. The entire PMD area (AP –2.46 ~ –2.7 mm, total 3 sections per animal) was imaged and quantified. Batch image processing of signal deconvolution was performed prior to automatic cell counting using the ZEN 2.6 blue software (Zeiss) where c-Fos positive cells were identified as filled objects with circularity values ( > 0.6) supplemented with visual confirmation of individual particles and size. For the confirmation of AAV infusion and fiber implantation sites, brain tissues were sectioned as described above and stained for GFP (chicken anti-GFP, 1:1,000 in 0.1% PBS-T, Abcam, ab 13970; Alexa Fluor 488-conjugated donkey anti-chicken secondary antibody, 1:1000 in 0.1% PBS-T).

## Statistical analysis

All statistical analyses were performed using GraphPad Prism (GraphPad Software). In behavioral experiments, a (two-tailed) unpaired Student's t-test were generally used, but two-way repeated-measures ANOVA (2-WAY RM ANOVA) was employed in the RTPA analysis with treatment groups (GFP vs. ChR2) and stimulation frequency (6 and 20 Hz) as a between-subjects factor and time as a within-subjects factor. For HPC terminals quantification, one-way ANOVA was used with the post hoc analysis of Dunett multiple comparison test. For secondary predator odor context conditioning, one-way repeated

ANOVA was used and followed by Dunnett's multiple comparison test. Where appropriate, two-way RM ANOVAs were followed by planned pairwise comparisons such as Sidak's multiple comparison. Two-way ANOVA were followed by Sidak's multiple comparison to compare the effects of testing 6 or 20 Hz in optostimulation studies. A simple linear regression analysis was used to detect the relationship between the dose-dependent change in investigation time vs. freezing. A non-linear fit was used to model the change in speed and head angle vs. the normalized distance from the shelter in US evoked shelter directed escape. Significance was defined as *$p < 0.05$, **$p < 0.01$, ***$p < 0.001$, ****$p < 0.0001$.

## Additional information

### Funding

| Funder | Grant reference number | Author |
|---|---|---|
| Canadian Institute of Health Research | 507489 | Junchul Kim |
| NSERC Discovery | 506730 | Junchul Kim |

The funders had no role in study design, data collection and interpretation, or the decision to submit the work for publication.

### Author contributions

Jee Yoon Bang, Conceptualization, Data curation, Formal analysis, Investigation, Project administration, Validation, Visualization, Writing – original draft, Writing – review and editing; Julia Kathryn Sunstrum, Danielle Garand, Data curation, Formal analysis, Investigation; Gustavo Morrone Parfitt, Investigation; Melanie Woodin, Funding acquisition, Resources, Writing – review and editing; Wataru Inoue, Data curation, Formal analysis, Investigation, Resources, Writing – review and editing; Junchul Kim, Conceptualization, Formal analysis, Funding acquisition, Project administration, Resources, Supervision, Writing – original draft, Writing – review and editing

### Author ORCIDs

Jee Yoon Bang http://orcid.org/0000-0003-2031-1642
Gustavo Morrone Parfitt http://orcid.org/0000-0003-0168-4099
Melanie Woodin http://orcid.org/0000-0003-2984-8630
Wataru Inoue http://orcid.org/0000-0002-2438-5123
Junchul Kim http://orcid.org/0000-0001-9920-3307

### Ethics

All procedures were approved by the Local Animal Care Committee (LACC) at University of Toronto. AUP2011332.

### Decision letter and Author response

Decision letter https://doi.org/10.7554/eLife.74736.sa1
Author response https://doi.org/10.7554/eLife.74736.sa2

## Additional files

### Supplementary files
• Transparent reporting form

### Data availability

Numerical data used to generate Figures 1-8 and Figure supplements are provided in the Figure Source Data files that correspond to figure labels. Custom written MATLAB code is uploaded on Zenodo. (https://doi.org/10.5281/zenodo.5899428).

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
