## [Editor Report]

In this study, the authors provide novel insights into the mechanisms by which the brain computes contextual information associated with innate threats in mice. Specifically, it provides the first causal evidence of a hippocampus-anterior hypothalamic pathway mediating spatial fear memory of ethological threats. Overall, these findings should interest a broad scientific audience.

---

## [Decision Letter]

**Decision letter after peer review:**

Thank you for submitting your article "Hippocampal-hypothalamic circuit controls context-dependent innate defensive responses" for consideration by *eLife*. Your article has been reviewed by 2 peer reviewers, and the evaluation has been overseen by a Reviewing Editor and Kate Wassum as the Senior Editor. The following individuals involved in review of your submission have agreed to reveal their identity: Bianca Silva (Reviewer #1); Douglas S Engelke (Reviewer #2).

Essential revisions:

While individual assessments and recommendations from each of the reviewers are included below, here we provide you with a brief list of items that we collectively consider to be essential revisions that must be addressed in order for the manuscript to be considered further for publication at *eLife*. In addition to addressing these essential revisions, please also address the individual points raised by each one of the reviewers in the public reviews and recommendations for authors, as we consider that addressing them will strengthen the paper.

1) Both reviewers agree that a revision of statistical analyses is required. For example, t-tests have been performed in cases in which ANOVA is the appropriate statistical test (e.g. data on Figure 8M). We encourage the authors to revise this and to adjust the accompanying statements in the text.

2) An extended description of the methods section of the paper is required. Currently, aspects of the methods employed by the authors are not entirely clear.

3) The authors should provide a more comprehensive discussion of their findings and consider them in light of other recent reports (e.g. Wang et a., 2021; Vale et al., 2020; Masferrer et al., 2020).

4) Please make sure the sex of the subjects is described in the abstract and methods and the N and sex distribution for each experiment is clear.

*Reviewer #1 (Recommendations for the authors):*

1) Continuous optogenetic stimulation for 2 minutes at 20Hz can cause neuronal damage or rebound effects. In figure 3 the authors show continuous 15 Hz stimulation for no more than 5 s and even within this short stimulation protocol, evoked responses seem to decrease over time (Figure 3 F). An ex-vivo control of prolonged stimulation efficacy would help clarify this issue.

2) Similarly, optogenetic terminal inhibition has proven challenging and ex-vivo confirmation of efficacy would be preferred. cFos data go in the right direction, but provide no temporal resolved information.

3) No information on how cFos image analysis was performed is provided. This should be explained in detail in the methods. How many sections per animal were analyzed? Was the cFos counting performed manually or automatically? Did the optic fiber impair tissue quality?

4) In Figure 6J and 8M, pairwise t-tests are not appropriate and authors should perform a TWO-WAY ANOVA.

I performed this statistical analysis from the submitted source data file for Figure 8M and the only significant difference is, when properly correcting for multiple comparison, VMH control+shelter vs no shelter, and no significant differences arise in AH. The authors should revise this part of the results.

Also, in Figure 8M it is not clear what the authors mean by "GFP-AHN vs. ArchT-AHN (unpaired t-test, two-tailed, t=7.005, df=6, ***p=0.0004)." This comparison should be better explained in the legend.

5) At line 60 in the introduction session, none of the references (22-25) refers to the second part of the sentence: "while its inhibition reduces defensive responses to predator threats". References for this part should be included.

6) At line 55 the recently discovered role of the PMD in context specific innate escape behavior should be cited (Wang et al., Neuron 2021)

7) In Extended Figure 1A it would be better to include a better representative picture of AH somatic ChR2 expression. Why does it look so different than any other inset? In addition, authors should show an example of "minimal spread to neighbouring hypothalamic areas" by showing a non-cropped image.

8) In the Results section the authors basically only refer to Extended data Figure 2I. The behavioral effect observed by low frequency stimulation and all the other results in the figure are very interesting and in line with the rest of the manuscript and should be detailed in the results.

9) In Figure 1 E-K the number of animals per condition is only stated in some panels. It should be specified in every panel.

10) I wonder why the authors chose to photostimulate HPC fibers in the AH at 15Hz in the ex vivo preparation, while the behavioral experiments are always performed at 6 or 20Hz.

11) Scale bar in Figure 8L PMD is weirdly displayed.

12) At line 268 a reference is needed

13) In the Discussion section, the authors should elaborate on how their study complements the retrosplenial cortexSC pathway mediating shelter-directed escape described in Vale et al., 2020 (https://doi.org/10.1101/2020.05.26.117598)

14) Despite not being directly targeted by the hippocampus, the VMH also directly modulates targeted escape (Masferrer et al., 2020). This should be mentioned in the discussion.

*Reviewer #2 (Recommendations for the authors):*

1) It is a bit confusing why the authors sometimes show 6 and 20hz stimulation in the main figures and sometimes only 20Hz in the main figure and 6Hz in supplemental. It is not clear whether the authors have a specific hypothesis for testing different frequencies of stimulations in their models. It is known that hypothalamus has frequency-dependent roles in behavior (Kunwar, 2015 – *eLife*; Lee, 2014 – Nature), but I found no mention or discussion about this in the manuscript.

2) Although authors identify with dashed lines the anatomy of the histochemistry presented, it is missing the anatomical landmarks that allow the readers to identify the boundaries of the regions as well as improve the ability to interpret the data. In Figure 6i it is impossible to have an anatomical reference of the hypothalamic areas. For example, putting a third ventricle region and other anatomical landmarks will benefit this and all other histology of the paper.

3) Figure 5 – not clear why the authors used ANOVA and T-test in different figures. If the comparison between the groups includes the different frequencies, ANOVA must be performed in all figures.

4) I would represent the freezing levels in percentage given that every test has a different duration. Otherwise it is hard to interpret across tests.

Other comments:

1) Authors should avoid "* = p<0.05", "** = p<0.01", etc. If the significance level established by the paper is 95%, a p < 0.001 does not make the data "more different" then a p=0.049. This is a common assumption of the field that we need to start removing from papers. I would stick with one single symbol.

2) Abstract:

(2a) Stating the work was done in mice can be useful for the readers.

(2b) in line 33.. "(AHN) is best positioned to perform this task"…, it reads a little odd that a brain area is positioned to perform the task. In fact, the mouse is performing the task and the structure may be processing, or controlling, a specific behavior.

3) Line 73-74: although I understand the authors mean that HPC project solely to AHN at the hypothalamic area, it would be better rephrasing, otherwise a non-expert reader could be misinformed and think that the HPC only project to AHN, which is not the case.

4) Not clear what the author means by struggle index (Figure 1K and 4I), missing detail of how this index was calculated.

5) Figure 1 – it would be very useful for the readers having a schematic cartoon with the timeline and workflow of the experiments considering the total time of all the tests and manipulations combined.

6) Figure 3 – ANOVA was performed in a N of 2, which doesn't make sense given you need variance, that should require at least 3 samples. In fact, no parametric stats should be run for N below 6 per group.

7) Figure 4C, no mention that 6 Hz was used.

8) Figure 4, labels are too small.

9) Figure 8K – are the levels of freezing being reported correct? IF yes, this is a spurious difference 0.5 seconds of freezing versus 1.5 sec it is absolutely irrelevant at the behavioral level. Same is true for Figure ext data 10, very low levels of freezing even in control animals, not really possible to interpret much with these levels of freezing.

10) Figure Extended data 9 – missing freezing levels during conditioning and distance traveled.

11) Authors should review the quality of the images of the paper as well as font size of the axis. There are several mismatching in size as well as blurry images that compromise the quality of the paper, e.g Figure 3E.

12) Methods:

(12a) animals – it seems that animals used also Jackson lab mice, but only Charles rivers source is cited

(12b) the description of the predator odor conditioning is confusing and lacking information. The authors say "Two different testing paradigms were employed", but they never explicitly tell , or ex, "paradigm 1" : …; instead it is mentioned AM/PM conditioning. For me it seems a lab internal jargon, I believe that a clear separation of both procedures would make it clearer to the reader to follow the procedure and replicate it for further advancement of the field. Also a clear explanation of why these two different protocols were performed is missing.

(12c) No mention of the source of L-felinine.

(12d) Authors should report the freezing quantification criteria, were the freezing hand scored in all the tasks? If yes, describe how authors blind the measures. IF was automatic, describe the features of freezing detection, e.g threshold for considering freezing.

[Editors’ note: further revisions were suggested prior to acceptance, as described below.]

Thank you for resubmitting your work entitled "Hippocampal-hypothalamic circuit controls context-dependent innate defensive responses" for further consideration by *eLife*. Your revised article has been evaluated by Kate Wassum (Senior Editor) and a Reviewing Editor.

The manuscript has been improved but there are some remaining issues that need to be addressed, as outlined below:

This paper will be of interest to neuroscientists, particularly those studying defensive behaviors. The authors provide novel insights into the mechanisms by which the brain computes contextual information associated with innate threats in mice. The experimental approach and data analysis are mostly adequate and the study provides the first causal evidence of a hippocampus-anterior hypothalamic pathway mediating spatial fear memory of ethological threats. The implementation of more robust statistical tests, as well as more detailed Methods and Discussion sections, should serve to strengthen an already elegant study.

Essential revisions:

The reviewers agree that the authors have addressed most of their initial comments and that the paper is greatly improved. However, we request that the authors incorporate the reviewers' suggestions regarding: (1) statistical reporting (Reviewer 1), and (2) data presentation (Reviewer 2).

Please also make sure that the statistical reporting includes full reporting of the results, e.g., for 2-way ANOVA both interaction and main effects.

Please include a key resource table.

*Reviewer #2 (Recommendations for the authors):*

I acknowledge the authors improved the methods section and therefore the quality of the manuscript. The discussion was enriched and provide much more insights about the new results presented in the manuscript on the light of the current literature. In general the authors put efforts to review the major points.

1 -Regarding the individual data (individual dots per animal) the overcrowding of the figure can be solved place some level of transparency in the dots. I still insist that having the individual dots for each animal is crucial for transparency and interpretation of the data of the entire paper.

---

## [Author Response]

1) Both reviewers agree that a revision of statistical analyses is required. For example, t-tests have been performed in cases in which ANOVA is the appropriate statistical test (e.g. data on Figure 8M). We encourage the authors to revise this and to adjust the accompanying statements in the text.

We thank the editors for encouraging us to revise the statistical test. We have now replaced t-tests with ANONA and adjusted the accompanying statements in the results and figure legends.

2) An extended description of the methods section of the paper is required. Currently, aspects of the methods employed by the authors are not entirely clear.

We have revised and added further details in the methods section for clarity. Please see our responses to reviewers’ comments for more details.

3) The authors should provide a more comprehensive discussion of their findings and consider them in light of other recent reports (e.g. Wang et a., 2021; Vale et al., 2020; Masferrer et al., 2020).

We agree that a more comprehensive discussion would significantly improve the manuscript. As suggested, we have updated the Discussion section to better integrate our findings in light of other recent reports (Wang et a., 2021; Vale et al., 2020; Masferrer et al., 2020).

4) Please make sure the sex of the subjects is described in the abstract and methods and the N and sex distribution for each experiment is clear.

Thank you for the suggestion. The information (the N size and sex of the subjects) has been included in the abstract and methods accordingly.

Reviewer #1 (Recommendations for the authors):1) Continuous optogenetic stimulation for 2 minutes at 20Hz can cause neuronal damage or rebound effects. In figure 3 the authors show continuous 15 Hz stimulation for no more than 5 s and even within this short stimulation protocol, evoked responses seem to decrease over time (Figure 3 F). An ex-vivo control of prolonged stimulation efficacy would help clarify this issue.

While we cannot completely exclude its possibility, our postmortem tissue analysis did not show any noticeable neuronal damage in the AHN area after optogenetic stimulation experiments. The potential rebound effect (observed often upon continuous optogenetic inhibition) and stimulation efficacy of continuous optogenetic stimulation can indeed be measured in ex-vivo electrophysiology experiments. To clarify whether these issues would impact the behavioural effects of optostimulation, however, one must ultimately conduct an extensive series of in vivo unit recording experiments with freely moving mice, which we believe is beyond the scope of the current work.

2) Similarly, optogenetic terminal inhibition has proven challenging and ex-vivo confirmation of efficacy would be preferred. cFos data go in the right direction, but provide no temporal resolved information.

We agree with the reviewer and fully acknowledge the well-known challenge for demonstrating the efficacy of optogenetic terminal inhibition. While our c-Fos data does not offer a high temporal resolution to tease out exactly when the terminal activity is suppressed or released from the inhibition, we believe it still provides the overall efficacy of optogenetic terminal inhibition. Similar to the above issues associated with continuous optostimulation, the temporal dynamics of optoinhibtion effects must ultimately be addressed by in vivo unit recording experiments with freely moving mice.

3) No information on how cFos image analysis was performed is provided. This should be explained in detail in the methods. How many sections per animal were analyzed? Was the cFos counting performed manually or automatically? Did the optic fiber impair tissue quality?

Thank you for your suggestion. The method section has been revised to add the missing details relevant to this concern.

4) In Figure 6J and 8M, pairwise t-tests are not appropriate and authors should perform a TWO-WAY ANOVA.I performed this statistical analysis from the submitted source data file for Figure 8M and the only significant difference is, when properly correcting for multiple comparison, VMH control+shelter vs no shelter, and no significant differences arise in AH. The authors should revise this part of the results.Also, in Figure 8M it is not clear what the authors mean by "GFP-AHN vs. ArchT-AHN (unpaired t-test, two-tailed, t=7.005, df=6, ***p=0.0004)." This comparison should be better explained in the legend.

Thank you for your corrections. We have replaced the t-tests with ANOVA tests in Figure 8M. Please note that one-way ANOVA was chosen first to test how the escape-induced AHN activity changes upon HPC-AHN pathway inhibition and shelter removal, which revealed a significant effect for both treatments; one-way ANOVA is justified because the opto-inhibition targets specifically the AHN, not VMHdm or PMD. A subsequent Two-way ANOVA, including all data for AHN, VMHdm, and PMD, revealed no significant interaction between the treatments and the brain regions, indicating that the treatment effect does not depend on the brain areas. Considering these three brain areas are heavily connected to each other, this is not surprising.

5) At line 60 in the introduction session, none of the references (22-25) refers to the second part of the sentence: "while its inhibition reduces defensive responses to predator threats". References for this part should be included.

We have revised the texts and references accordingly.

6) At line 55 the recently discovered role of the PMD in context specific innate escape behavior should be cited (Wang et al., Neuron 2021)

We have revised the texts and references accordingly.

7) In Extended Figure 1A it would be better to include a better representative picture of AH somatic ChR2 expression. Why does it look so different than any other inset? In addition, authors should show an example of "minimal spread to neighbouring hypothalamic areas" by showing a non-cropped image.

We have updated the figure accordingly.

8) In the Results section the authors basically only refer to Extended data Figure 2I. The behavioral effect observed by low frequency stimulation and all the other results in the figure are very interesting and in line with the rest of the manuscript and should be detailed in the results.

We have updated the result section accordingly.

9) In Figure 1 E-K the number of animals per condition is only stated in some panels. It should be specified in every panel.

We have updated the figures accordingly.

10) I wonder why the authors chose to photostimulate HPC fibers in the AH at 15Hz in the ex vivo preparation, while the behavioral experiments are always performed at 6 or 20Hz.

The ex vivo electrophysiology experiment was guided by a previous study from Dyu Lin’s lab (Wang et al., 2015 Neuron PMID: 25754823) and conducted long before we started the behavioural experiments. We later found that 20 Hz stimulation evokes the most robust behavioural effects.

11) Scale bar in Figure 8L PMD is weirdly displayed.

We have updated the figures accordingly.

12) At line 268 a reference is needed

We have updated the reference accordingly.

13) In the Discussion section, the authors should elaborate on how their study complements the retrosplenial cortexSC pathway mediating shelter-directed escape described in Vale et al., 2020 (https://doi.org/10.1101/2020.05.26.117598)

Thank you for the insights. We have updated the Discussion section accordingly to elaborate on how our work and the findings in Vale et al., complement each other. It is plausible that the medial hypothalamic defense system and the RSP-SC pathway project to the same postsynaptic cells in the dorsal periaqueductal gray to support an organized escape to shelter. The HPC-AHN pathway may control the motivational drive to escape based on shelter availability while the RSP-SC pathway controls the escape direction based on shelter location.

14) Despite not being directly targeted by the hippocampus, the VMH also directly modulates targeted escape (Masferrer et al., 2020). This should be mentioned in the discussion.

We have updated the Discussion section which now includes the role of VMH in escape responses revealed by a recent unit recording experiment by Masferrer et al.,

Reviewer #2 (Recommendations for the authors):1) It is a bit confusing why the authors sometimes show 6 and 20hz stimulation in the main figures and sometimes only 20Hz in the main figure and 6Hz in supplemental. It is not clear whether the authors have a specific hypothesis for testing different frequencies of stimulations in their models. It is known that hypothalamus has frequency-dependent roles in behavior (Kunwar, 2015 – eLife; Lee, 2014 – Nature), but I found no mention or discussion about this in the manuscript.

We decided to show only 20 Hz data in Figure 1 to focus on the most important message that AHN stimulation induces escape-associated behaviours. After we described the frequency-dependent effects of AHN stimulation, we included both 6 and 20 Hz data in all of the following main figures (Figures 2, 4, and 5). To further clarify this intention and the frequency-dependent effects of AHN stimulation, we have revised the result section accordingly.

2) Although authors identify with dashed lines the anatomy of the histochemistry presented, it is missing the anatomical landmarks that allow the readers to identify the boundaries of the regions as well as improve the ability to interpret the data. In Figure 6i it is impossible to have an anatomical reference of the hypothalamic areas. For example, putting a third ventricle region and other anatomical landmarks will benefit this and all other histology of the paper.

Thank you for the suggestion. We have updated the figures accordingly.

3) Figure 5 – not clear why the authors used ANOVA and T-test in different figures. If the comparison between the groups includes the different frequencies, ANOVA must be performed in all figures.

Thank you for your corrections. We have replaced the t-tests with ANOVA in all cases (Figure 2, 4, and 5) where we compare the effects of 6 and 20 Hz frequency stimulations in GFP and ChR2 groups.

4) I would represent the freezing levels in percentage given that every test has a different duration. Otherwise it is hard to interpret across tests.

We have updated the figures accordingly.

Other comments:1) Authors should avoid "* = p<0.05", "**= p<0.01", etc. If the significance level established by the paper is 95%, a p < 0.001 does not make the data "more different" then a p=0.049. This is a common assumption of the field that we need to start removing from papers. I would stick with one single symbol.

We acknowledge a need for the proposed change and would be willing to adopt it. If *eLife* policy is in line with the reviewer’s suggestion, we will change the format.

(2) Abstract:(2a) Stating the work was done in mice can be useful for the readers.(2b) in line 33.. "(AHN) is best positioned to perform this task"…, it reads a little odd that a brain area is positioned to perform the task. In fact, the mouse is performing the task and the structure may be processing, or controlling, a specific behavior.

We have revised the abstract accordingly.

3) Line 73-74: although I understand the authors mean that HPC project solely to AHN at the hypothalamic area, it would be better rephrasing, otherwise a non-expert reader could be misinformed and think that the HPC only project to AHN, which is not the case.

We have revised the abstract accordingly.

4) Not clear what the author means by struggle index (Figure 1K and 4I), missing detail of how this index was calculated.

We have updated the method section to clarify how we measured and calculated the struggle index.

5) Figure 1 – it would be very useful for the readers having a schematic cartoon with the timeline and workflow of the experiments considering the total time of all the tests and manipulations combined.

Timelines were indeed used in the main figures involving multi-step procedures, but we are willing to add more if necessary.

6) Figure 3 – ANOVA was performed in a N of 2, which doesn't make sense given you need variance, that should require at least 3 samples. In fact, no parametric stats should be run for N below 6 per group.

While N of 2 mice were used for the experiment, we analyzed 7-8 slices per brain area listed in Figure 3. We have revised the method and figure legends to further clarify this.

7) Figure 4C, no mention that 6 Hz was used.

We have revised the figure accordingly.

8) Figure 4, labels are too small.

We have revised the figure accordingly.

9) Figure 8K – are the levels of freezing being reported correct? IF yes, this is a spurious difference 0.5 seconds of freezing versus 1.5 sec it is absolutely irrelevant at the behavioral level. Same is true for Figure ext data 10, very low levels of freezing even in control animals, not really possible to interpret much with these levels of freezing.

Please note that we quantified the freezing level during only 9 seconds of the ultrasound presentation window. 2.5 seconds of freezing shown in control animals (Extended Data Figure 10) is equivalent to 28% of freezing.

10) Figure Extended data 9 – missing freezing levels during conditioning and distance traveled.

While we do have data for freezing levels and distance travelled during conditioning, we intentionally omit them to avoid a redundancy and focus more on the behavioural changes observed during retrieval phase. If it is critical to show the data, we are willing to include them.

11) Authors should review the quality of the images of the paper as well as font size of the axis. There are several mismatching in size as well as blurry images that compromise the quality of the paper, e.g Figure 3E.

We have revised the figure accordingly.

12) Methods:(12a) animals – it seems that animals used also Jackson lab mice, but only Charles rivers source is cited

We have revised the method section accordingly.

(12b) the description of the predator odor conditioning is confusing and lacking information. The authors say "Two different testing paradigms were employed", but they never explicitly tell , or ex, "paradigm 1" : …; instead it is mentioned AM/PM conditioning. For me it seems a lab internal jargon, I believe that a clear separation of both procedures would make it clearer to the reader to follow the procedure and replicate it for further advancement of the field. Also a clear explanation of why these two different protocols were performed is missing.

Thank you for the suggestion. We have indicated Paradigm 1 (P1) and Paradigm 2 (P2) indicating the AM/PM conditioning and the conditioning paradigm that shows development of the predator-cue associated context avoidance learning. Further explanations for the two different protocols were added.

(12c) No mention of the source of L-felinine.

We have revised the method section accordingly.

(12d) Authors should report the freezing quantification criteria, were the freezing hand scored in all the tasks? If yes, describe how authors blind the measures. IF was automatic, describe the features of freezing detection, e.g threshold for considering freezing.

We have added details into the methods to describe how freezing was manually quantified in a treatment-blind manner.

[Editors’ note: further revisions were suggested prior to acceptance, as described below.]

Essential revisions:The reviewers agree that the authors have addressed most of their initial comments and that the paper is greatly improved. However, we request that the authors incorporate the reviewers' suggestions regarding: (1) statistical reporting (Reviewer 1),

We agree with the feedback and have ensured that there is full statistical reporting (e.g., for 2-WAY ANOVA both interaction and main effects) in all figure legends to improve our manuscript.

and (2) data presentation (Reviewer 2).

Thank you for the suggestion. We have now updated all figure panels with column bar graphs to show individual data.

Please also make sure that the statistical reporting includes full reporting of the results, e.g., for 2-way ANOVA both interaction and main effects.

Please see the response (1).

Please include a key resource table.

We have now included a key resource table. It can be found at the beginning of “Materials and methods” section.

Reviewer #2 (Recommendations for the authors):I acknowledge the authors improved the methods section and therefore the quality of the manuscript. The discussion was enriched and provide much more insights about the new results presented in the manuscript on the light of the current literature. In general the authors put efforts to review the major points.1 -Regarding the individual data (individual dots per animal) the overcrowding of the figure can be solved place some level of transparency in the dots. I still insist that having the individual dots for each animal is crucial for transparency and interpretation of the data of the entire paper.

Though initially concerned with overcrowding, we strongly concur with this recommendation and insights. We have now included individual dots.